# Light sets the brain's daily clock by regional quickening and slowing of the molecular clockworks at dawn and dusk

Suil Kim[1], Douglas G McMahon[1,2]*

[1]Vanderbilt Brain Institute, Vanderbilt University, Nashville, United States;
[2]Department of Biological Sciences, Vanderbilt University, Nashville, United States

**Abstract** How daily clocks in the brain are set by light to local environmental time and encode the seasons is not fully understood. The suprachiasmatic nucleus (SCN) is a central circadian clock in mammals that orchestrates physiology and behavior in tune with daily and seasonal light cycles. Here, we have found that optogenetically simulated light input to explanted mouse SCN changes the waveform of the molecular clockworks from sinusoids in free-running conditions to highly asymmetrical shapes with accelerated synthetic (rising) phases and extended degradative (falling) phases marking clock advances and delays at simulated dawn and dusk. Daily waveform changes arise under ex vivo entrainment to simulated winter and summer photoperiods, and to non-24 hr periods. Ex vivo SCN imaging further suggests that acute waveform shifts are greatest in the ventro-lateral SCN, while period effects are greatest in the dorsomedial SCN. Thus, circadian entrainment is encoded by SCN clock gene waveform changes that arise from spatiotemporally distinct intrinsic responses within the SCN neural network.

## Editor's evaluation

The work by Kim and McMahon describes a useful ex vivo approach to address fundamental circadian mechanisms. Their strategy allows optogenetic manipulation of suprachiasmatic nucleus (SCN) slices with red-light, in order to mimic light input. By conducting classic entrainment protocols they are able to correlate data obtained in whole animals, with ex-vivo changes in core-clock gene expression (PER2LUC). Thus, they assess how some key organismal circadian properties, related to light entrainment, can be recapitulated ex-vivo in isolated SCN slices, while other ones may depend on more complex organismal physiological intricate networks. The major conclusions of the work include: rapid optogenetic activation of SCN neurons results in changes in the SCN PER2 waveform that correlate with changes in behavior induced by seasonal or non-24h environmental cues.

*For correspondence: douglas.g.mcmahon@vanderbilt. edu

**Competing interest:** The authors declare that no competing interests exist.

## Introduction

A key biological function of circadian clocks is to encode local environmental time, and the seasons, through interactions with the daily light cycle. Most organisms live in a rhythmic environment where daily environmental changes occur corresponding with solar time, and their endogenous 24 hr timing mechanism, or the circadian clock, enables adjustment of their physiology and behavior accordingly. In mammals, the central clock—the suprachiasmatic nucleus (SCN) of the hypothalamus—represents solar time to synchronize peripheral tissue clocks in the rest of the body, and it drives the expression of daily and seasonal behaviors in tune with the temporal structure of the environment.

Classical behavioral studies performed by manipulating light cycles have revealed fundamental principles in how circadian rhythms are reset and synchronized by external time cues at the level of

behavioral outputs (*Pittendrigh and Daan, 1976a*; *Pittendrigh and Daan, 1976b*), but the molecular basis of clock setting and synchronization remains to be fully explained. Entrainment or alignment of circadian locomotor behavior with light cycles is achieved based on differential sensitivity of the circadian rhythm to the timing of light exposure – phase delays result from light exposure in the early circadian night, phase advances from light in the late night, and there is a 'dead zone' in the mid-day where light does not reset the rhythm (*Pittendrigh and Daan, 1976b*). At the molecular level, the basis of the mammalian circadian clock is self-sustained circadian oscillations of core clock genes arranged in an autoregulatory transcription-translation feedback loop, with transcription factors *Clock* and *Bmal1* driving the circadian expression of *Period* (*Per*) and *Cryptochrome* (*Cry*) genes that mediate negative feedback within the clockworks (*Takahashi, 2017*). Light stimulation results in acute induction of *Per1/2* (*Takahashi, 2017*). Conventional time-point clock gene expression profiling from animals under different light cycles has provided a snapshot of how the clock gene rhythms in the SCN are influenced by various lighting conditions (*Messager et al., 2000*; *Schwartz et al., 2011*), but with limited temporal resolution that does not fully capture dynamic changes in the clock gene rhythms induced by different light cycles.

The advent of clock gene reporters (e.g. *Per1*::GFP, PER2::LUC) (*Kuhlman et al., 2000*; *Yoo et al., 2004*) enabled assaying the motion of the circadian clock in real time and with low variability. However, to date the fundamental question of how clock genes in the SCN encode different lighting conditions has been primarily approached by manipulating light exposure in vivo and subsequently explanting the SCN into slice culture in constant darkness to retrospectively infer the in vivo entrained state, due to technical challenges in mimicking retinal light input ex vivo. While much progress has been made with this paradigm, this approach can have significant limitations. The SCN activity observed in a free-running condition reflects relaxation of the explanted SCN network back toward baseline, rather than active encoding of entrainment (*Rohr et al., 2019*), and explantation can disrupt expression of the in vivo state (*Pendergast et al., 2009*).

Here, combining long-term organotypic explant culture, cyclic red light optogenetic stimulation, and the PER2 bioluminescent reporter, we assess how the clock gene rhythms in the ex vivo SCN change in real time to achieve entrainment to light cycles. We have instituted an ex vivo optogenetic experimental system that provides precise timing, duration, and intensity of recurring input stimulation to the explanted SCN while tracking clock gene rhythms at high temporal resolution. Acute optogenetic channelrhodopsin-2 (ChR2) stimulation of SCN neurons with 470 nm blue light to make them fire at light-driven spike frequency (e.g. 8–15 Hz, *Jones et al., 2015*; *Mazuski et al., 2018*), as does retinal light input, has been shown to be effective to reset circadian rhythms in vivo and ex vivo. However, long-term blue light illumination in culture (without opsins) results in phototoxicity, reducing cell viability (470 nm, 1 Hz, *Stockley et al., 2017*) and degrading many biological processes, including cell growth (470 nm, continuous, *Ohara et al., 2002*) and respiration (425~500 nm, continuous, *Robertson et al., 2013*). Notably, red light is more tolerable in vitro (488 nm vs 558 or 640 nm, continuous, *Wäldchen et al., 2015*) and in vivo (440~500 nm vs. 540~660 or 655~695 nm, continuous, *De Magalhaes Filho et al., 2018*). Recent development of optogenetic actuators responding to red light (*Klapoetke et al., 2014*; *Lin et al., 2013*) prompted us to test whether red light can be utilized for long-term optogenetic stimulation, and to study how core clock gene PER2 rhythms in the ex vivo SCN are dynamically altered and reset by repeated light stimulation to achieve light entrainment.

Here, we uncover that PER2 rhythms in the ex vivo SCN under entrainment to optogenetic light cycles show contraction of the rising phase and elongation of the falling phase depending on the timing of light exposure, and reveal ex vivo SCN plasticity at the clock gene level similar to canonical features of light-induced plasticity in circadian behavior. Aspects of circadian plasticity to light entrainment and regional heterogeneity of light responsiveness are apparently intrinsic to the SCN clockworks.

## Results
### An integrated system for long-term optogenetic stimulation and bioluminescence recording

To entrain the SCN slice with optogenetic stimulation, light pulses must be given periodically for multiple days to weeks. While repeated and long-term optogenetic stimulation of the SCN with ChR2

has been used to successfully entrain circadian locomotor behavior (*Jones et al., 2015*; *Tackenberg et al., 2021*), long-term blue light exposure in culture can decrease cell viability via toxic byproducts (*Stockley et al., 2017*). To test the effects of sustained blue light exposure on SCN slices, we delivered blue light pulses (470 nm, 1.2 mW/mm$^2$) for 12 hr to SCN slices expressing a bioluminescent translational reporter of the core clock gene *Per2*, PER2::LUC (*Yoo et al., 2004*), but no optogenetic construct. We applied 10 Hz, 10 ms light pulses that can be used to optogenetically drive SCN neurons to fire at light-driven spike frequency in vivo (*Jones et al., 2015*; *Mazuski et al., 2018*; *Meijer et al., 1998*). We found that PER2::LUC bioluminescence became arrhythmic following the prolonged blue light exposure (*Figure 1A and B*). This effect was not reversible with a medium change (*Figure 1A*), suggesting that long-term blue light exposure per se can impair circadian rhythmicity in SCN slice cultures.

Since side effects of light exposure decrease with increasing irradiation wavelength (*Tyssowski and Gray, 2019*; *Wäldchen et al., 2015*), we tested whether using red light (625 nm) mitigates light impairment of SCN rhythms. Twelve hr red light pulses (625 nm, 10 Hz, 10 ms, 1.25 mW/mm$^2$) did not significantly affect the PER2::LUC rhythm in SCN slices (*Figure 1A and B*), suggesting that using red light stimulation could be more feasible for prolonged optogenetic light stimulation ex vivo. We thus expressed a red light-activated opsin, ChrimsonR, fused with a red fluorescent protein tdTomato (tdT) throughout SCN slices using synapsin promoter-controlled AAVs targeting all SCN neurons (AAV-Syn-ChrimsonR-tdT, *Klapoetke et al., 2014*) to mimic widespread retinal photic inputs to the SCN (*Chen et al., 2011*; *Fernandez et al., 2016*; *Figure 1C*).

To extend the duration over which we can observe the SCN slice throughout entrainment, we improved the quality and robustness of PER2::LUC rhythmicity by using brain slices from younger mice that usually survive longer in culture (*Humpel, 2015*), and by using stabilized glutamine media that were shown to reduce ammonia production and improve cell viability in cell culture (*Christie and Butler, 1999*; *Imamoto et al., 2013*; *Figure 1—figure supplement 1*). With these modifications, ex vivo SCN rhythms were stable for more than three weeks, long enough to conduct entrainment paradigms without culture medium changes that may perturb ex vivo SCN rhythms and entrainment. For precise temporal control of optogenetic stimulation of multiple SCN slices, we integrated an optogenetic stimulation apparatus into a multi-channel luminometer (*Figure 1D*). To minimize the potential effect of LED-generated heat on SCN rhythmicity while achieving sufficient light intensity for optogenetic stimulation, we set up a light delivery path in which an LED placed outside the incubator housing the luminometer could deliver 625 nm light through a fiber optic cable and a collimation lens (*Figure 1D*). We then created a program interface for remotely operating PER2::LUC luminometry and optogenetic stimulation in a coordinated manner (*Figure 1—figure supplement 2*).

## Discrete optogenetic light pulses differentially alter SCN clock gene waveforms to induce phase and period responses

To test how this integrated system induces phase resetting of circadian rhythms that fundamentally underlies circadian entrainment, we applied 10 Hz optogenetic light stimulations to ChrimsonR-expressing SCN slices on the 3rd or 4th day of recording at three different circadian times (CT) as defined by the timing of the intrinsic PER2::LUC rhythm of each SCN slice. We chose 10 Hz stimulation as light-driven SCN spike frequency in vivo ranges from 7–13 Hz (*Meijer et al., 1998*). By convention, CT12 was defined as the peak time of the PER2::LUC rhythm and is correlated in vivo with the onset of nocturnal behavioral activity (*Yoo et al., 2004*). Thus, the rising phase (CT0–12) of the PER2::LUC rhythm represents the day phase of intrinsic SCN circadian time, while the falling phase (CT12–24) corresponds to physiological night. Light stimulation in vivo at CT6, CT14, and CT21 induces representative phase responses in the locomotor circadian rhythm (no phase shift, phase delays, and phase advances, respectively) (*Daan and Pittendrigh, 1976*; *Johnson, 1999*). We found that the phase responses to ChrimsonR-mediated stimulation in ex vivo SCN indeed mimicked the phase responses of circadian behavior in intact mice to light stimulations (*Figure 2A and C*). Optogenetic stimulation of SCN slices at CT6 induced little phase shift (0.14 ± 0.19 hr; mean ± SEM), whereas optogenetic stimulation at CT14 and CT21 induced significant phase delays and advances, respectively (CT14: –4.25 ± 0.76 hr, CT21: 4.38 ± 0.67 hr; *Figure 2C*).

Strikingly, optogenetic stimulation of SCN slices differentially altered the waveform of PER2::LUC rhythms on the stimulated cycle, depending on the timing of the stimulation (*Figure 2B*). Stimulation

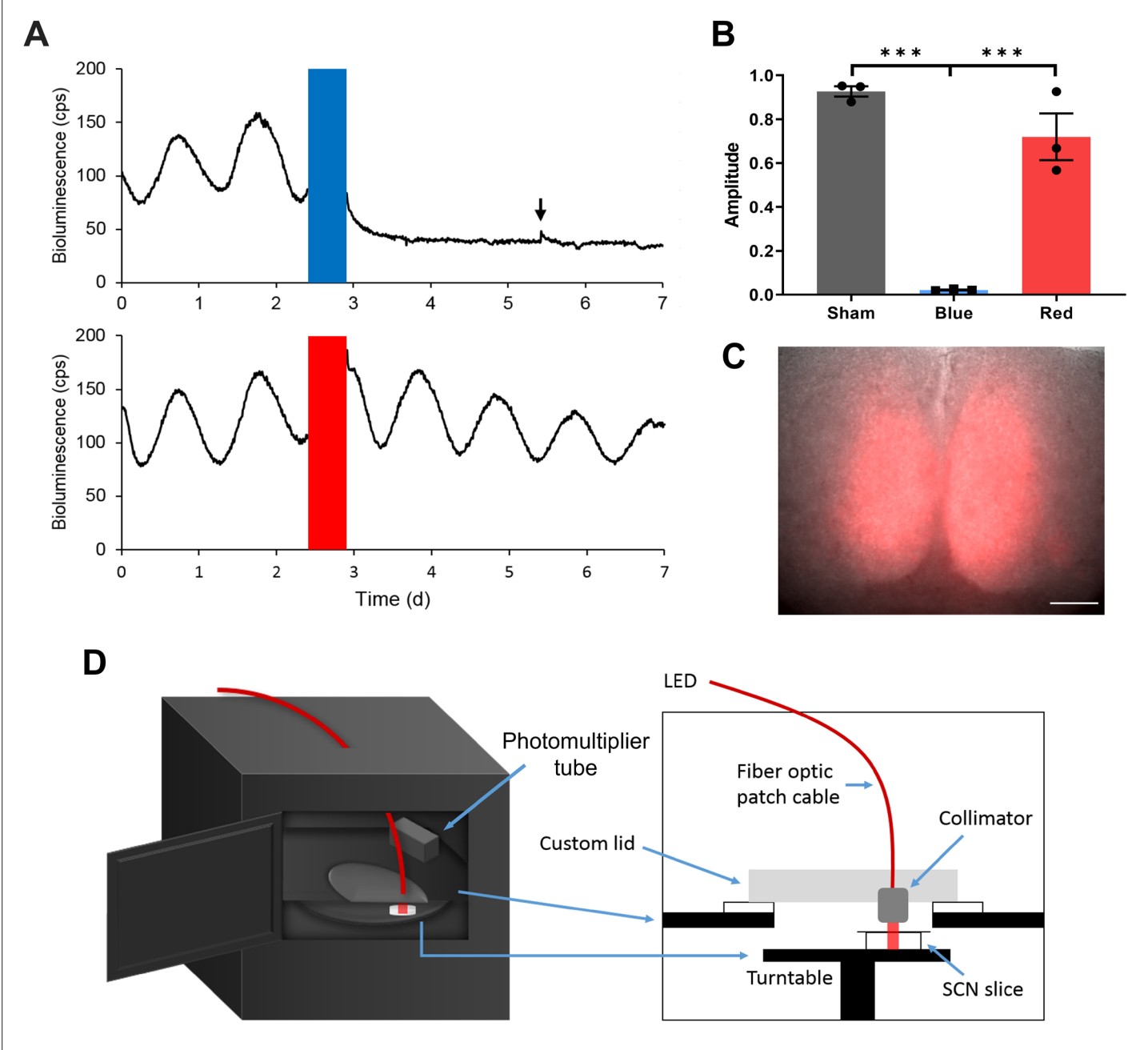

**Figure 1.** Long-term Optogenetic Stimulation System for Circadian Entrainment Ex Vivo. (**A**) Representative PER2::LUC bioluminescence rhythms of adult SCN slices exposed to either red (top) or blue (bottom) 10 Hz light pulses (red or blue bars) for 12 hr. The black arrow indicates the timing of media change. (**B**) Fold change in the rhythm amplitude following sham, blue, or red light exposure (Student's t-test, mean ± SEM, n = 3, ***p < 0.001). (**C**) Merged ChrimsonR-tdT fluorescence and the brightfield images of an SCN slice. Scale = 100 μm. (**D**) Diagrams showing a multi-channel luminometer integrated with an optogenetic stimulation apparatus.

The online version of this article includes the following source data and figure supplement(s) for figure 1:

**Source data 1.** Source data for *Figure 1B*.

**Figure supplement 1.** Improved PER2::LUC rhythmicity in SCN slices explanted from young mice to culture medium containing stabilized glutamine.

**Figure supplement 2.** Diagram of an integrated system for long-term luminescence recording and optogenetic stimulation.

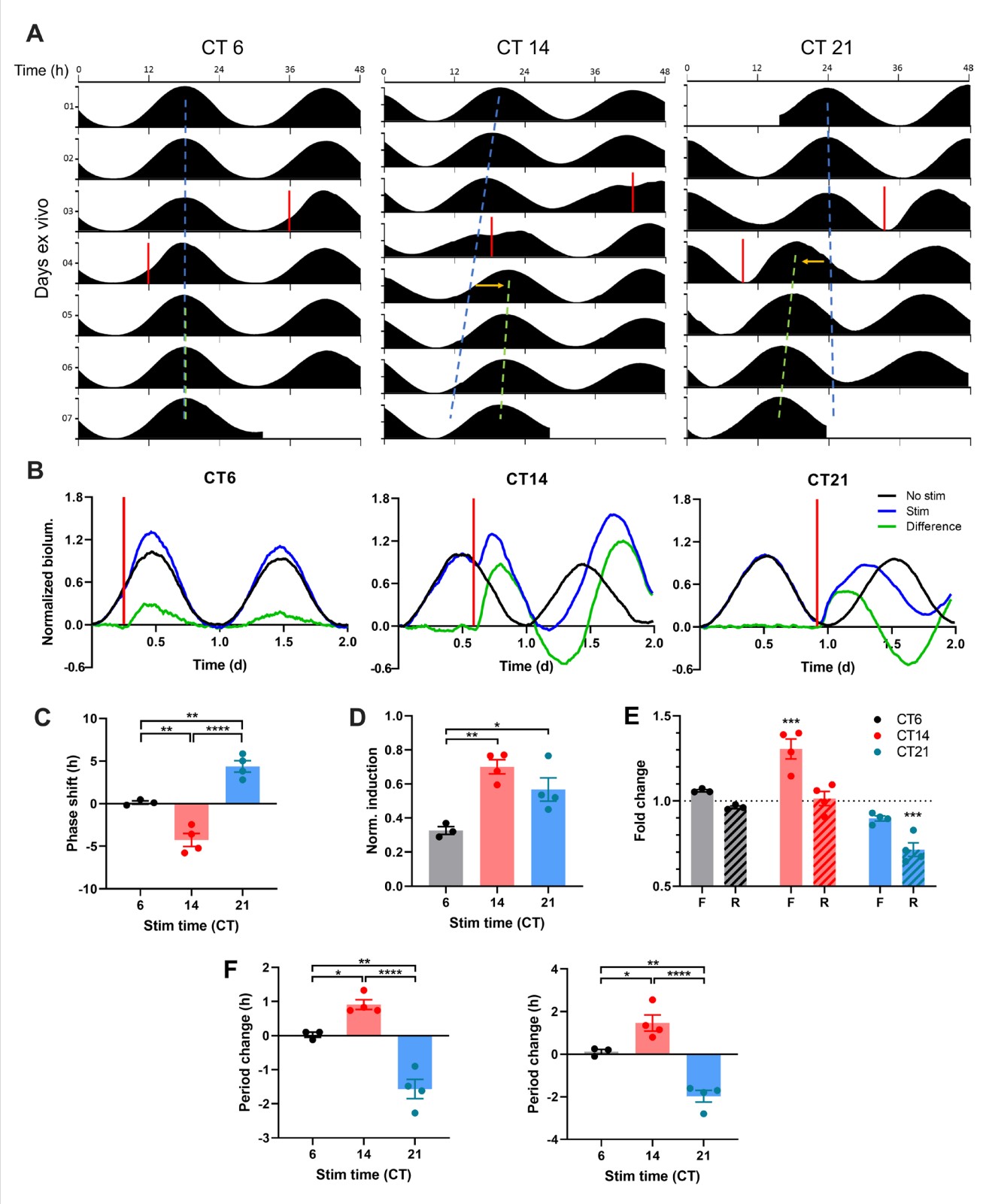

**Figure 2.** ChrimsonR-driven Optogenetic Stimulation Alters the Waveform of SCN PER2 Rhythms to Reset the SCN Clock. (**A**) Representative double-plotted PER2::LUC bioluminescence actograms of SCN slices stimulated with single 15 min 10 Hz optogenetic pulses (red bar) at CT6 (left), 14 (middle), and 21 (right). Linear regressions of the pre-stimulation and post-stimulation cycle peaks are indicated as the blue and green dashed lines, respectively. Phase shifts are depicted by yellow arrows. (**B**) Representative PER2::LUC rhythms of SCN slices before and after stimulation (red bar). Green traces

*Figure 2 continued on next page*

*Figure 2 continued*

depict differences in normalized bioluminescence between the pre-stim and post-stim rhythms (black and blue traces, respectively). (**C**) Quantification of phase shifts following stimulation. Positive and negative phase shifts indicate phase advance and delay, respectively. (One-way ANOVA with Tukey's multiple comparisons test, mean ± SEM, n = 3–4, **p < 0.01, ****p < 0.0001). (**D**) Normalized PER2::LUC induction following stimulation. (One-way ANOVA with Tukey's multiple comparisons test, mean ± SEM, n = 3–4, *p < 0.05, **p < 0.01). (**E**) Fold change in duration of the rising (R; dashed boxes) and falling (F; open boxes) phases of PER2::LUC rhythms following stimulation. (RM two-way ANOVA with Sidak's multiple comparisons tests, mean ± SEM, n = 3–4, ***p < 0.001). (**F**) Quantification of period changes following stimulation using linear regression of peaks (left) and Lomb-Scargle periodogram (right). Positive and negative period changes indicate period lengthening and shortening, respectively. (One-way ANOVA with Tukey's multiple comparisons test, mean ± SEM, n = 3–4, *p < 0.05, **p < 0.01, ****p < 0.0001).

The online version of this article includes the following source data for figure 2:

**Source data 1.** Source data for *Figure 2C–F*.

at night (CT14, CT21) induced greater acute increases in PER2::LUC expression in SCN slices than did stimulation in the day (CT6) (CT6: 0.33 ± 0.02 artificial unit (AU), CT14: 0.70 ± 0.04 AU, CT21: 0.57 ± 0.07 AU; *Figure 2D*). The differential PER2::LUC induction led to differential changes in the duration of specific phases of PER2::LUC rhythms (*Figure 2E*). Stimulation at CT14 induced an acute increase in PER2, elongating the falling phase (1.31 ± 0.06 fold change (FC)) to induce a phase delay, whereas stimulation at CT21 accelerated the rising phase (0.72 ± 0.04 FC) to induce a phase advance. Stimulation at CT6, however, did not significantly alter either the rising or the falling phase duration (rising phase: 0.97 ± 0.01 FC, falling phase: 1.06 ± 0.01 FC) despite small PER2::LUC induction, thus causing little phase shift. Together, differential waveform changes in PER2 rhythms underpin time-dependent phase responses to light stimulation in the SCN.

Further, the acute phase shifts were accompanied by a subsequent sustained changes in the free-running circadian period (i.e. circadian period of an oscillator in the absence of external time cues such as light) of SCN slices that persisted for multiple days as an after-effect of the acute phase-shifting stimulations (*Figure 2A and F*). Similar to after-effects of light induced phase shifts on circadian loco-motor behavior (*Pittendrigh and Daan, 1976a*), phase delays in PER2::LUC rhythms following stimulation at CT14 led to period lengthening on subsequent cycles, whereas phase advances following CT21 stimulation resulted in period shortening (linear regression method. CT14: 0.91 ± 0.14 hr, CT21: –1.57 ± 0.28 hr; Lomb-Scargle periodogram. CT14: 1.46 ± 0.38 hr, CT21: –1.98 ± 0.28 hr; *Figure 2F*). These after-effects on period were not the result of persistent waveform changes – the PER2::LUC waveform returned to its free-running sinusoidal form after the stimulated cycle. Thus, the SCN itself, in isolation from extra-SCN clocks and neural inputs, has plasticity of the circadian period following acute light stimulation.

## Daily phase-specific waveform changes drive entrainment at the clock gene level

Individual phase shifts by external time cues are the building blocks of circadian entrainment (*Pitten-drigh and Daan, 1976b*). Daily, repeated phase shifts to light adjust the endogenous circadian period to match the 24 hr day and align circadian rhythms in a particular temporal relationship with the daily light-dark cycle (i.e. phase angle of entrainment). Remarkably, animals can entrain to single light pulses given repeatedly as a cycle (T-cycle), even to cycle periods that deviate modestly from 24 hr, such as a 22 hr light cycle (*Pittendrigh and Daan, 1976b*). This so-called one-pulse entrain-ment is the simplest form of entrainment well-documented in the literature and can be used to easily interpret whether biological clocks can entrain to certain external cues. To test whether optogenetic stimulation can entrain the isolated SCN clock, we delivered periodic optogenetic light stimulation at intervals (optogenetic day-night cycle lengths, or T-cycles) that significantly deviated from the near-24 hr endogenous period of the SCN, to clearly differentiate the entrained state from free-running. One cohort of SCN slices was stimulated every 22 hr (T22) while the other cohort was stimulated every 25 hr (T25, *Figure 3A*). SCN in both cohorts demonstrated the canonical responses defining circadian entrainment—matching of clock period to the period of the input stimulus (*Figure 3B*, *Figure 3— figure supplements 1 and 2*), adopting a stable timing relationship with the repeating stimulus (stable phase angle, *Figure 3C*, *Figure 3—figure supplement 3*), and initiating a subsequent free-run from the point in time of the entrained phase angle upon cessation of the stimulus (*Figure 3A*). Impor-tantly, PER2::LUC rhythms in the SCN entrained to the different T-cycles with different phase angles of

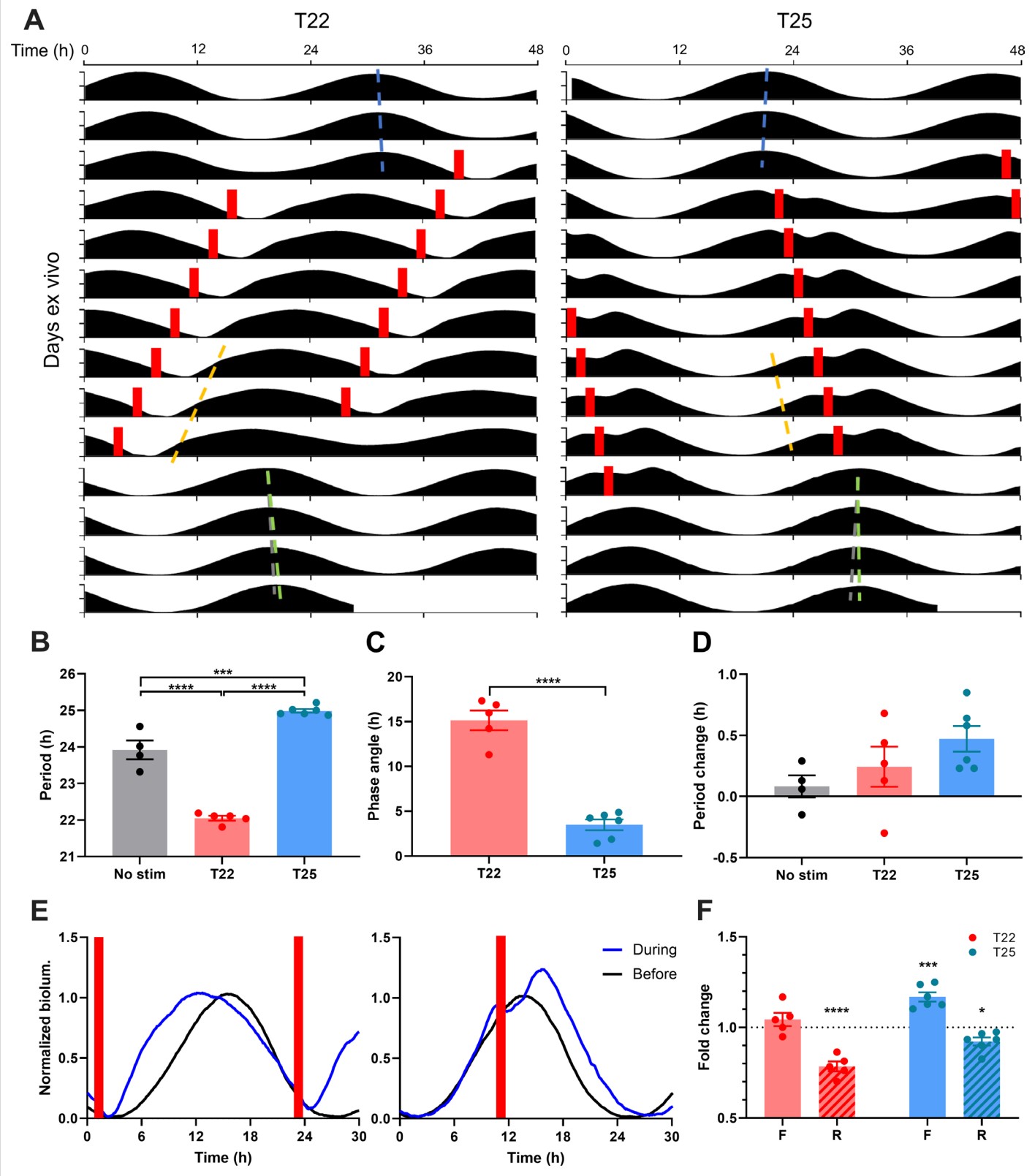

**Figure 3.** Optogenetic Cycles Entrain PER2::LUC Rhythms in the SCN by Triggering Daily Waveform Changes that Match the SCN Clock to the Cycle Period. (**A**) Representative double-plotted PER2::LUC bioluminescence actograms of SCN slices entrained with 1–1.5 hr 10 Hz optogenetic pulse (red bars) every 22 hr (left) or 25 hr (right). Linear regressions of the pre- and post-entrainment cycle peaks are indicated as the blue and green dashed lines, respectively. Yellow dashed lines indicate half-maxes on the rising phase during entrainment. Grey dashed lines indicate the pre-entrainment cycle

*Figure 3 continued on next page*

*Figure 3 continued*

period as a reference. (**B–D**) Quantification of period during entrainment (**B**), phase angle of entrainment (**C**), period change by entrainment (**D**). (**B and D**) were analyzed using one-way ANOVA with Tukey's multiple comparisons tests (mean ± SEM, n = 4–6, ****p < 0.0001, ***p < 0.001), (**C**) was analyzed using Student's t-test (mean ± SEM, n = 5–6, ****p < 0.0001). (**E**) Representative waveforms of PER2::LUC bioluminescence rhythms before (black trace) and during (blue trace) entrainment to T22 (left) and T25 cycles (right). Red bars depict optogenetic stimulation during entrainment. (**F**) Fold change in duration of the rising (R; dashed boxes) and falling (F; open boxes) phases of PER2::LUC rhythms during entrainment compared with before entrainment. (RM two-way ANOVA with Sidak's multiple comparisons tests, mean ± SEM, n = 5–6, ***p < 0.001, ****p < 0.0001).

The online version of this article includes the following source data and figure supplement(s) for figure 3:

**Source data 1.** Source data for *Figure 3B–D and F*.

**Figure supplement 1.** PER2::LUC rhythms in SCN slices entrain to optogenetic T-cycles.

**Figure supplement 2.** Quantification of period during entrainment (left) and period changes following entrainment (right) using Lomb-Scargle periodogram.

**Figure supplement 2—source data 1.** Source data for *Figure 3—figure supplement 2*.

**Figure supplement 3.** Acrophase fitting of PER2::LUC bioluminescence actograms from *Figure 3A*.

entrainment (*Figure 3C*, *Figure 3—figure supplement 3*), as predicted by the non-parametric model of circadian entrainment, and previously observed at the level of behavioral outputs (*Pittendrigh and Daan, 1976b*). PER2::LUC rhythms under T22 and T25 cycles showed that the periodic stimulation was aligned with the late falling and the late rising phases, respectively, where acute stimulation produces phase advances and delays, respectively (*Figure 3A*). Indeed, PER2::LUC rhythms in the SCN were entrained to short and long period cycles by daily phase advances or daily phase delays, respectively. The daily phase advances and delays were respectively derived from repeated acceleration of the PER2::LUC rising phase (0.79 ± 0.03 fold change) or elongation of the falling phase (1.17 ± 0.03 fold change; *Figure 3A, E and F*). Taken together, our results demonstrate that PER2::LUC rhythms in SCN slices can entrain to periodic optogenetic stimulation, and reveal that PER2 rhythms in the SCN encode different input cycle lengths via repeated rising phase shortening or falling phase lengthening.

We also tested whether PER2 rhythms in SCN slices show plasticity of endogenous clock period in constant darkness following T-cycle entrainment. T-cycle entrainment by repeated light pulses in vivo produces after-effects on the period of circadian locomotive behavior: short and long T-cycles produce period shortening and lengthening, respectively (*Pittendrigh and Daan, 1976b*; *Schwartz et al., 2011*). As phase shifts by single stimulation had profound period after-effects on PER2::LUC rhythms (*Figure 2D*), repeated phase shifts by periodic stimulation were expected to have significant period after-effects. Surprisingly, however, repeated phase advances in PER2::LUC rhythms in SCN slices during T22 entrainment and phase delays during T25 entrainment did not produce statistically significant period after-effects (*Figure 3D*, *Figure 3—figure supplement 2*). This suggests that phase shifts (rather than changes in the endogenous period) are a primary driver of matching circadian PER2 rhythms in the SCN to the period of T-cycles, as proposed by non-parametric model of entrainment (*Pittendrigh and Daan, 1976b*).

## The SCN clock entrains to skeleton photoperiods with a minimum tolerable night

Circadian oscillators in the SCN encode the length and timing of the daily light period (i.e. photoperiod) (*Goldman, 2001*; *Rusak and Morin, 1976*), thereby promoting seasonal changes in physiology and behavior. Strikingly, light cycles consisting of only brief light pulses defining dawn and dusk (i.e. skeleton photoperiods) have been shown to simulate most aspects of full photoperiods at the behavioral level, with the interval between the brief dawn and dusk pulses determining the photoperiodic state of circadian behavior (*Pittendrigh and Daan, 1976b*). This remarkable ability of the circadian system to lock onto the timing of light transitions can, in principle, result from properties of circadian photoreception in the retina (input), downstream behavioral modulation (output), or properties of the SCN clock itself. Here we have tested directly whether the SCN clock itself can be entrained to different photoperiods by brief daily transitions that simulate dawn and dusk.

We applied to SCN slices an optogenetic equivalent of skeleton photoperiods mimicking short (winter-like), equinox, or long (summer-like) photoperiods (8 hr, 12 hr, and 16 hr daylight per day),

respectively (*Figure 4A and B*). For the 12:12 skeleton photoperiod entrainment, we gave short opto-
genetic stimulations twice per day (12 hr apart). We initiated entraining stimulations with one stimu-
lation at the trough of the free-running PER2::LUC rhythm to mimic dawn, and the other stimulation
at the peak to mimic dusk (*Figure 4A,B*, 12:12). The 12:12 skeleton indeed entrained SCN slices
such that PER2::LUC rhythm period during entrainment became matched to the 24 hr optogenetic
light cycle length (*Figure 4C*, *Figure 4—figure supplement 1*), and the phase angle of entrainment
(measured as the difference in time between the dusk pulse and the half-max on the rising phase of
PER2::LUC rhythm) was stable (*Figure 4D*, *Figure 4—figure supplement 2*). To test for 8:16 short day
skeleton photoperiod entrainment, we initiated twice-daily optogenetic stimulations 8 hr apart, with
the 'dawn' pulse given 2 hr after the trough of the free-running PER2::LUC rhythm, and the 'dusk' pulse
given 2 hr before the peak (*Figure 4A,B*, 8:16). The 8:16 skeleton photoperiod entrained PER2::LUC
rhythms to 24 hr period (*Figure 4C*, *Figure 4—figure supplement 1*) with a stable phase angle
(*Figure 4D*, *Figure 4—figure supplement 2*) such that the rising phase (SCN day) was encompassed
within the 8 hr 'short day' interval between the 'dawn' and 'dusk' stimulations (*Figure 4B*). To test
for entrainment to 16:8 long day skeleton photoperiods, we initiated optogenetic stimulations twice
daily 16 hr apart, with the 'dawn' stimulation applied 2 hr before the trough and the 'dusk' stimulation
applied 2 hr after the peak (*Figure 4A and B*). In contrast to the 12:12 and the 8:16 skeletons, when
SCN were presented 16:8 skeleton photoperiods they did not maintain alignment of the PER2::LUC
rising phase (SCN day) with the 'daytime' interval (16 hr), but instead rapidly phase-advanced across
the 'dawn' pulse until achieving similar phase angles to the 8:16 skeleton, stably aligning the SCN
day (rising) phase within the 8 hr short day interval (*Figure 4B–D*). This phenomenon in SCN slices
replicates the phase jump of circadian locomotor behavior in rodents (*Pittendrigh and Daan, 1976b*;
*Figure 4—figure supplement 3*) and flies (*Pittendrigh and Minis, 1964*) during attempted entrain-
ment to long-day skeleton photoperiods, a characteristic expression of circadian plasticity in which
entrainment to short-day skeleton photoperiods is more stable. Together, our results show that the
SCN clock itself can entrain to different photoperiods by dawn and dusk light pulses, and suggest
that the phase-jumping behavior during skeleton periods is derived from SCN-intrinsic clock plasticity.

Entrainment to different skeleton photoperiods also altered the molecular waveform of the SCN
clock (*Figure 4F*). Stimulations at dawn shortened the PER2::LUC rising phase (8:16 skeleton: 0.84
± 0.02 fold change (FC), 12:12 skeleton: 0.88 ± 0.03 FC, 16:8 skeleton: 0.70 ± 0.03 FC), while stim-
ulations at dusk lengthened the falling phase (8:16 skeleton: 1.16 ± 0.04 FC, 12:12 skeleton: 1.15 ±
0.03 FC, 16:8 skeleton: 1.30 ± 0.03 FC), to match the clock period to 24 hr and set the phase angle
of entrainment (*Figure 4F and G*). The net effect was that PER2 rhythms show asymmetric waveforms
with the falling phase longer than the rising phase in all skeleton photoperiods (*Figure 4F and G*).
However, changes in the waveform did not persist when the rhythms were free-running in constant
darkness following entrainment (*Figure 4G*), indicating that the waveform changes are direct effects
of optogenetic light stimulation accelerating and decelerating different phases of the molecular clock
during entrainment. Additionally, as entrainment to 8:16 skeleton photoperiods causes period length-
ening of circadian behavior in vivo (*Pittendrigh and Daan, 1976b*), it induced a trend toward period
lengthening of PER2::LUC rhythms as an after-effect (0.47 ± 0.11 hr), although these did not reach
statistical significance (*Figure 4E*, *Figure 4—figure supplement 2*). Taken together, skeleton photo-
periods entrain PER2 rhythms in the SCN via opposing actions of light pulses on the duration of the
rising and the falling phases to match the clock period to 24 hr.

To determine the waveform of PER2::LUC rhythms in SCN stably entrained to a long skeleton
photoperiod, we entrained SCN slices to skeleton photoperiods whose daytime interval was gradu-
ally extended from 12 hr to 14 hr to 16 hr (*Figure 5A*). All 9 SCN slices tested entrained to the 12:12
skeleton photoperiod, and most (8 out of 9) maintained entrainment when the skeleton was expanded
to 14:10, while only a few (2 out of 9) maintained entrainment when the skeleton was expanded to
16:8, and most (7 out of 9) executed a phase jump to the 8:16 phase angle (*Figure 5A and C*). This is
consistent with previous findings in fruit flies and mice that a phase jump in circadian behavior is more
likely to occur as the daytime interval of skeleton photoperiods gets longer than 12 hr, a phenomenon
called the 'minimum tolerable night' (*Pittendrigh and Daan, 1976b*; *Stephan, 1983*).

SCN slices that did stably entrain to the 16:8 long skeleton photoperiod exhibited a broader
PER2::LUC rhythm waveform with more prominent PER2::LUC inductions following stimulation at
each dawn and dusk transition, compared to the same slices under the 12:12 skeleton photoperiod

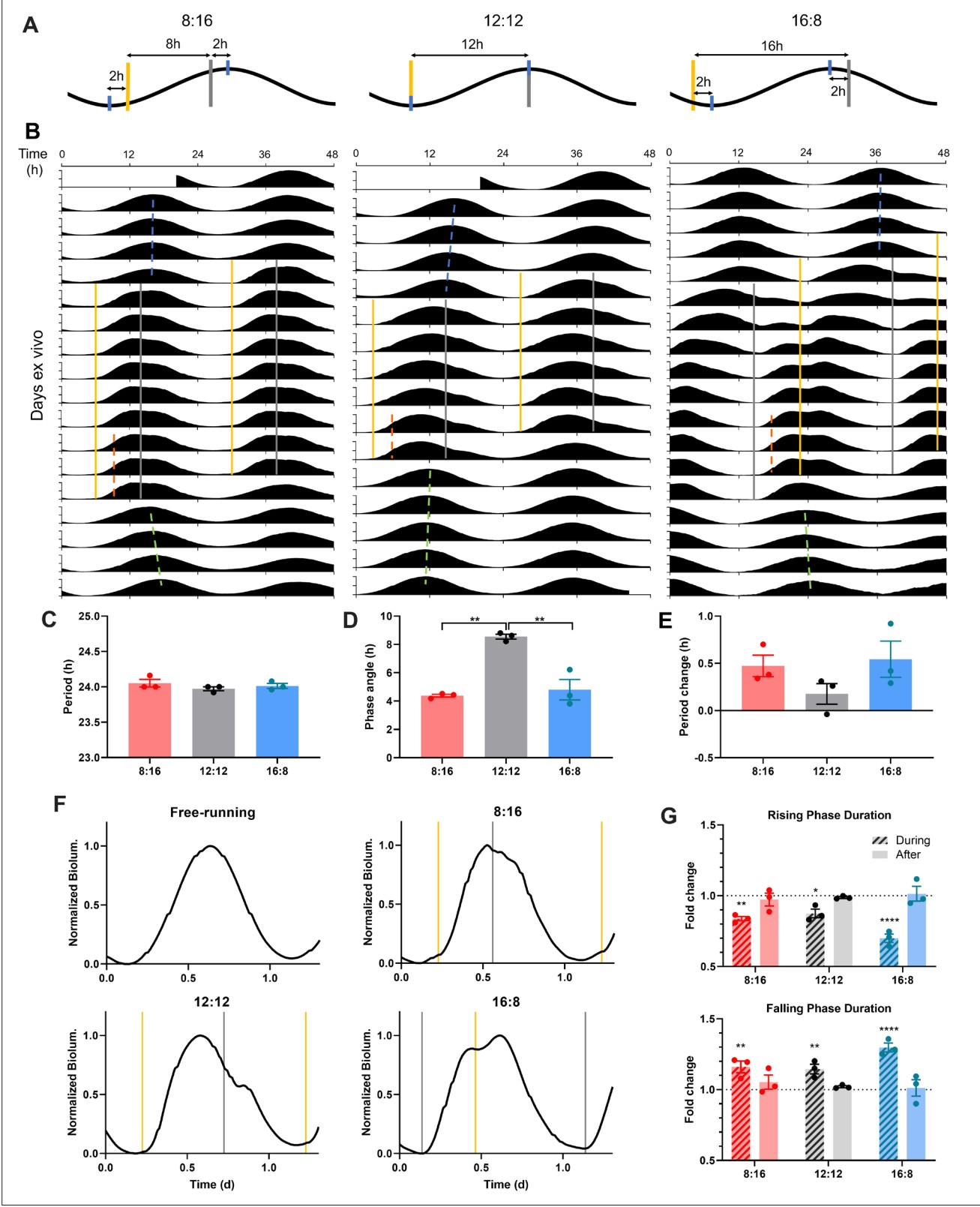

**Figure 4.** SCN entrain to optogenetic simulation of dawn and dusk via differential PER2::LUC rhythm waveform changes. (**A**) Schematic diagram of optogenetic stimulation paradigm for 8:16 (left), 12:12 (middle), and 16:8 (right) skeleton photoperiod entrainment. Optogenetic pulses (yellow and grey bars for nominal dawn and dusk pulses, respectively) were given twice (8 hr, 12 hr, or 16 hr apart) every 24 hr, targeting near the peak and trough (blue ticks) of PER2::LUC rhythms (black). (**B**) Representative double-plotted PER2::LUC bioluminescence actograms of SCN slices entrained with

*Figure 4 continued on next page*

*Figure 4 continued*

15 min 10 Hz red optogenetic pulses at 8:16 (left), 12:12 (middle), or 16:8 (right) interval of 24 hr. Linear regressions of the pre-, post-entrainment cycle peaks and during-entrainment cycle half-maxes are indicated as the blue, green, and orange dashed lines, respectively. (C–E) Quantification of period during entrainment (C), phase angle of entrainment (D), period change following entrainment (E). (One-way ANOVA with Tukey's multiple comparisons test, mean ± SEM, n = 3, **p < 0.01). (F) Representative waveforms of PER2::LUC rhythms in SCN slices in a free-running condition (before and after entrainment), and those entrained to optogenetic 8:16, 12:12, 16:8 photoperiod entrainment. Optogenetic light pulses were given at times indicated by yellow and grey lines marking nominal dawns and dusks, respectively. (G) Fold changes in the duration of the rising phase (top) and the falling phase (bottom) during and after entrainment, compared to before entrainment. (RM two-way ANOVA with Sidak's multiple comparisons tests, mean ± SEM, n = 3, *p < 0.05, **p < 0.01, ****p < 0.0001).

The online version of this article includes the following source data and figure supplement(s) for figure 4:

**Source data 1.** Source data for *Figure 4C–E and G*.

**Figure supplement 1.** Acrophase fitting of PER2::LUC bioluminescence actograms from *Figure 4B*.

**Figure supplement 2.** Quantification of (A) period during entrainment and (B) period changes following entrainment using Lomb-Scargle periodogram.

**Figure supplement 2—source data 1.** Source data for *Figure 4—figure supplement 2*.

**Figure supplement 3.** Phase jump in mouse circadian behavior during long skeleton photoperiod entrainment.

(*Figure 5B*). Providing SCN slices with simulation of dawn and dusk transitions was sufficient to mimic waveform broadening of the clock gene rhythms induced by complete long photoperiods (e.g. a cycle of 16 hr light and 8 hr darkness) (*Messager et al., 1999*; *Schaap et al., 2003*). Dawn stimulation for the 16:8 skeleton aligned with the late falling phase of the rhythms and led to a greater increase in the slope of the rising phase than did the 12:12 dawn stimulation aligned with the trough (*Figure 5B*). At dusk, the timing of the 16:8 skeleton stimulation at the early falling phase was later than that of the 12:12 stimulation, triggering greater PER2::LUC induction (*Figure 5B*). Our results show that the timing of light-dark transitions at dawn and dusk differentially drives photoperiodic effects on the clock gene rhythm waveform in the SCN likely via the observed changes in the magnitude of *Per* induction.

## The SCN clock expresses regionally distinct clock-resetting capacities

The SCN clock is a functional network of heterogenous cellular oscillators which themselves are autonomous clocks. Recently, genetic dissection approaches (e.g. cell-type-specific knockouts) (*Mieda et al., 2015*) and real-time imaging of the ex vivo SCN (*Evans et al., 2013*) further revealed aspects SCN timekeeping at the network level, broadly defining the ventral region marked by vasoactive intestinal polypeptide (VIP) secreting neurons, and the dorsal region marked by arginine vasopressin (AVP) secreting neurons, as distinct functional nodes within the SCN network. SCN slices explanted from mice entrained to extreme lighting conditions such as 20:4 photoperiods (20 hr light:4 hr darkness) or 22 hr light cycles (11 hr light:11 hr darkness) showed regional phase differences and coupling profiles in PER2::LUC rhythms within the SCN distinct from those of SCN slices explanted from mice under a standard light cycle (12 hr light:12 hr darkness) (*Azzi et al., 2017*; *Evans et al., 2013*). However, these post-hoc ex vivo studies assay the relaxation of previous light cycle encoding, not the process of entrainment itself (*Rohr et al., 2019*), and it remains unclear how light input alters network function of the SCN clock in real time. As phase shifts in circadian rhythms by such light pulses underlie light entrainment (*Pittendrigh and Daan, 1976b*), we assessed how the circadian phase shifts by single light pulses impact the network state of the SCN clock. To do that, we combined optogenetic stimulation of the SCN ex vivo with spatially imaging real-time PER2::LUC bioluminescence in SCN using a light-tight microscope equipped with an LED illumination system.

Pan-neuronal optogenetic stimulation of SCN neurons was applied at CT14 or CT21 and the subsequent network-level changes in the SCN clock were observed following this phase-delaying or phase-advancing stimulation. Subregional bioluminescence rhythm analysis and subsequent hierarchical clustering analysis revealed the spatiotemporal distribution of circadian phase, period, phase response, and period response within the SCN (*Figure 6*). Before stimulation, the circadian oscillator network in SCN slices exhibited a baseline phase distribution where the dorsomedial lip was most phase-leading and the ventrolateral area the most phase-lagging, and the distribution of circadian

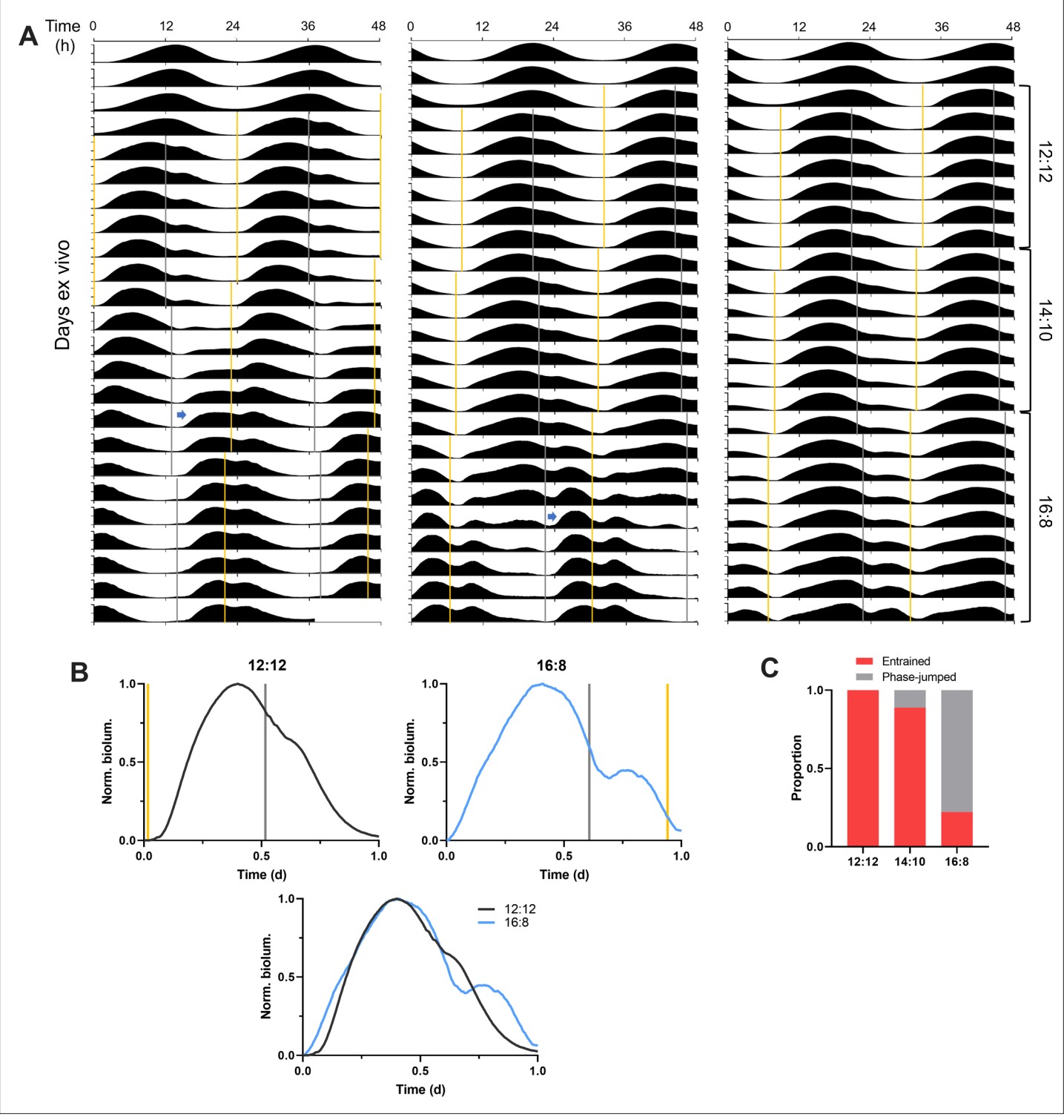

**Figure 5.** Gradual shifts from equinox to long optogenetic skeleton photoperiods reveal SCN PER2::LUC rhythms stably entrained to long skeleton photoperiods. (**A**) Representative double-plotted PER2::LUC bioluminescence actograms of SCN slices entrained to optogenetic skeleton photoperiods (yellow and gray bars for nominal dawn and dusk pulses, respectively) from the 12:12 skeleton to the 14:10 to the 16:8. The left and the middle actogram show that a phase jump (blue arrow) begins during the 14:10 and the 16:8 skeleton photoperiods, respectively. The right actogram depicts SCN PER2::LUC rhythms entrained to all the three skeleton photoperiods without a phase jump. (**B**) Representative waveforms of the PER2::LUC rhythms entrained to the 12:12 (upper left) and the 16:8 (upper right) skeleton photoperiods simulated by optogenetic stimulation (yellow and grey bars for nominal dawn and dusk pulses, respectively). The bottom plot shows a superimposed image of the two upper plots. (**C**) Proportion of entrained SCN

*Figure 5 continued on next page*

*Figure 5 continued*

slices (without a phase jump) and the phase-jumped slices for the 12:12, 14:10, and 16:8 skeleton photoperiods (n = 9).

The online version of this article includes the following source data for figure 5:

**Source data 1.** Source data for *Figure 5C*.

period was almost uniform across the SCN (*Figure 6A*). Optogenetic stimulation induced phase shifts and the period changes of different magnitudes in different regions of the SCN (*Figure 6A*).

Following CT14 stimulation, the lateral region showed larger phase delays than did the medial region. In contrast, period lengthening effects were more prominent in the medial SCN than in the lateral SCN, indicating an inverse correlation between phase change and period plasticity among SCN subregions (*Figure 6A and C*). To classify the SCN into subregions with different rhythmic properties in an unsupervised manner, we performed group-averaged clustering analysis and identified a lateral-medial axis with three clusters for phase shifts (lateral shell: −1.33 ± 0.18 hr, lateral core: −1.64 ± 0.17 hr, medial: −0.64 ± 0.30 hr) and a ventrolateral-dorsomedial axis with three clusters for period changes (dorsomedial: 0.36 ± 0.16 hr, ventrolateral: 0.08 ± 0.14 hr, in-between: 0.31 ± 0.14 hr; *Figure 6C and D*). In terms of changes in phase coupling, the phase variance within the SCN was increased following stimulation (*Figure 6B*), indicating a larger phase desynchrony in the SCN. The phase difference between the medial and the lateral regions increased following stimulation (pre-stim: 1.50 ± 0.11 hr, post-stim: 2.17 ± 0.25 hr) as larger phase delays occurred in the phase-lagging, lateral region than in the phase-leading, medial region (*Figure 6C and D*).

Following CT21 stimulation, the ventrolateral or ventral SCN showed larger phase advances and smaller period shortening than did the dorsomedial or dorsal SCN, indicating again an inverse correlation in magnitude between the phase shift and the period change among SCN subregions (*Figure 6A and C*). Cluster analysis identified a ventrolateral-dorsomedial axis with three clusters for phase shifts (ventrolateral shell: 2.49 ± 0.27 hr, ventrolateral core: 2.87 ± 0.29 hr, dorsomedial: 2.21 ± 0.22 hr) and a ventral-dorsal axis with three clusters for period changes (ventral shell: −0.69 ± 0.11 hr, ventral core: −0.55 ± 0.13 hr, dorsal: −0.83 ± 0.15 hr; *Figure 6C and E*). Also, the phase variance was increased following stimulation (*Figure 6B*) as the phase difference between the phase-leading and the phase-lagging clusters increased (pre-stim: 1.76 ± 0.19 hr, post-stim: 2.52 ± 0.22 hr; *Figure 6C and E*). This increased phase desynchrony can be explained by two phenomena: the ventral part of the phase-leading medial lip region showed large phase advances, while the dorsal part of the phase-lagging lateral region exhibited smaller phase advances (*Figure 6C*).

Taken together, the network-level changes in the circadian phase and period distributions within the SCN in response to discrete light pulses reveal that the lateral or ventrolateral SCN exhibits large phase shifts and small period changes, whereas the medial or dorsomedial SCN displays small phase shifts and large period changes. To test whether a higher regional expression of the ChrimsonR optogenetic construct in the lateral or ventrolateral SCN might be associated with the larger phase shifts of this region, we analyzed the ChrimsonR-tdT fluorescence intensity in different subregions corresponding to phase-shift clusters and compared with the magnitude of phase shifts. For both CT14 and CT21 stimulations, subregions exhibiting large phase shifts had lower levels of ChrimsonR-tdT expression (*Figure 6—figure supplement 1*), suggesting that regionally different phase responses are not directly derived from spatial ChrimsonR expression patterns. Given this distribution, and that we applied synchronous stimulation across the entire SCN, our results suggest that this subregional heterogeneity in resetting responses is intrinsic to the neurons or circuits in the ventrolateral and dorsomedial SCN.

## Discussion

In this study, we assessed at high-resolution the temporal and spatial dynamics of core clock gene rhythms in the ex vivo SCN following single optogenetic light exposures and various forms of optogenetic light cycles to gain further insights into light resetting and entrainment of clock gene rhythms in the SCN. Using real-time bioluminescent recording of PER2 expression combined with recurring optogenetic stimulation, we confirmed that the isolated SCN strikingly recapitulates many of the canonical features of circadian clock entrainment in intact animals, including (1) phase responses with period after-effects, (2) period matching and phase angle differences to stimuli of non-24-hour cycles, and

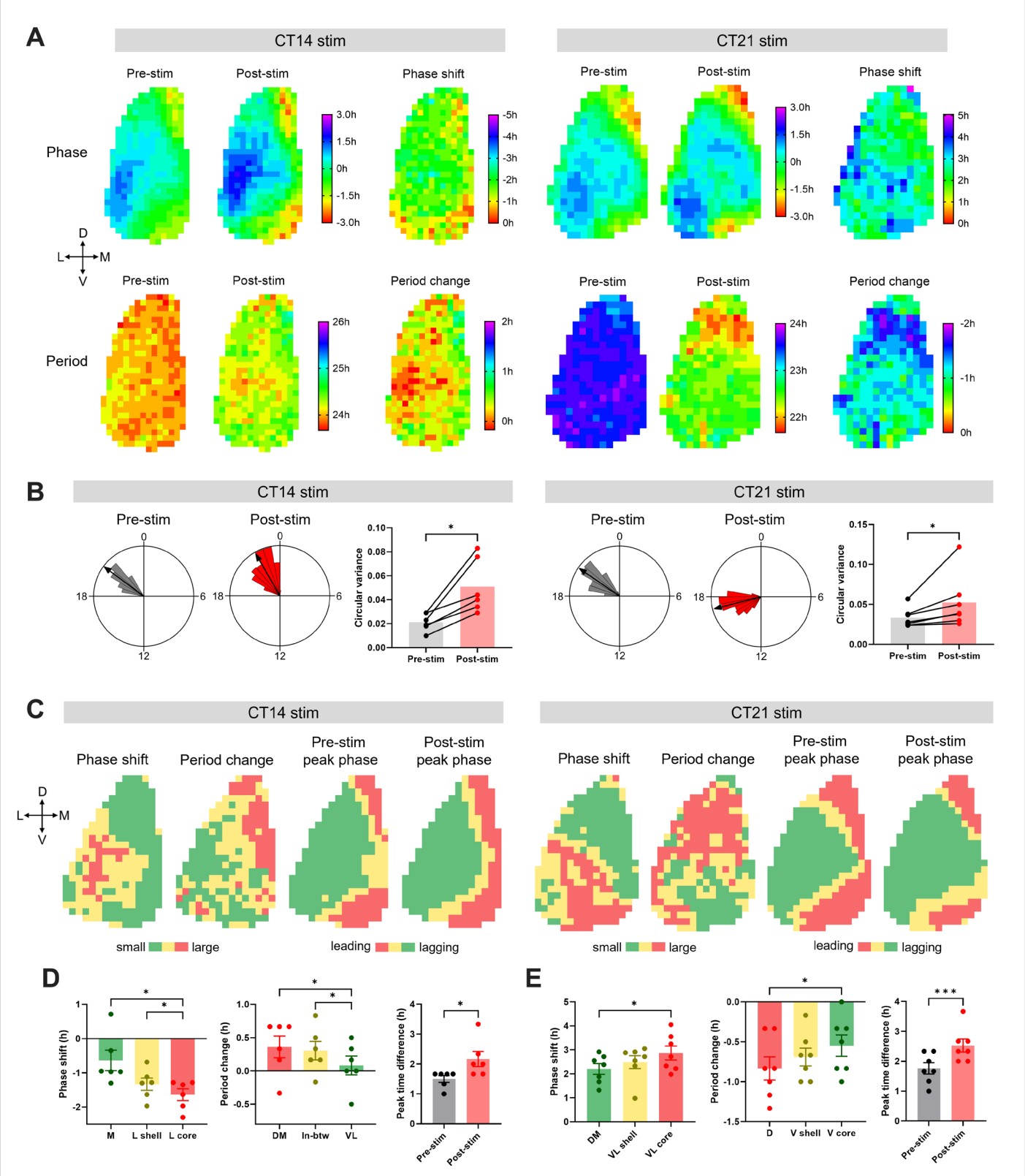

**Figure 6.** Regional distribution of circadian phase and period in the SCN, and heterogeneity of the phase and the period responses in the SCN to optogenetic light pulses. (**A**) The upper row shows phase maps of PER2::LUC bioluminescence rhythms in the unilateral SCN before and after optogenetic stimulation at CT14 (left) or CT21 (right), and regional distribution of the phase shifts in response to corresponding stimulation. The lower row displays the period maps before and after the stimulation, and the period change maps. The crossed arrows denote the SCN orientation. D =

*Figure 6 continued on next page*

*Figure 6 continued*

dorsal, V = ventral, M = medial, L = lateral. The phase maps depict peak times in hours relative to the mean in each condition. Pre-stim and post-stim phases indicate the last peak before stimulation and the first peak after stimulation, respectively. Positive and negative phase shifts indicate phase advances and delays, respectively. Positive and negative period changes indicate period lengthening and shortening, respectively. (**B**) Representative Rayleigh plots (24 hr circular plots) for regional peak PER2::LUC phases in the SCN, and bar graphs of the circular variance before and after optogenetic stimulation at CT14 (left) or CT21 (right). (Wilcoxon matched-pairs test, n = 6–7, *p < 0.05). The arrows in the Rayleigh plots indicate mean Rayleigh vectors. (**C**) Group-averaged clusters of the phase shift, period change, pre-stimulation peak phase, and post-stimulation peak phase for CT14 (left) or CT21 (right) stimulation (n = 6–7). Clusters were hierarchically formed using Ward's minimum variance method. Different colors indicate different clusters. The crossed arrows denote the SCN orientation. D = dorsal, V = ventral, M = medial, L = lateral. (**D–E**) Phase shifts (left) and period changes (middle) in corresponding clusters in (**C**), and peak time differences between the phase-leading and the phase-lagging clusters (right) for CT14 (**D**) or CT21 stimulation (**E**). Phase shifts and period changes were analyzed using RM One-way ANOVA with Tukey's multiple comparisons test, mean ± SEM, n = 6–7, *p < 0.05. Peak time differences were analyzed using paired t-test, mean ± SEM, n = 6–7, *p < 0.05, ***p < 0.001.

The online version of this article includes the following source data and figure supplement(s) for figure 6:

**Source data 1.** Source data for *Figure 6B and D–E*.

**Figure supplement 1.** ChrimsonR-tdT fluorescence levels in different phase shift clusters in the *Figure 6C* for (**A**) CT14 and (**B**) CT21 stimulation.

**Figure supplement 1—source data 1.** Source data for *Figure 6—figure supplement 1*.

(3) differential entrainment to skeleton photoperiods with a minimum tolerable night. Entrainment is expressed as transformation of the ongoing rhythmic waveform of the clock protein we monitored, PER2, from approximately sinusoidal in free-running conditions to highly asymmetrical trajectories with accelerated synthetic (rising) phases, and extended degradative (falling) phases of PER2 protein abundance mediating clock advances and delays, respectively. Spatiotemporal analysis revealed intrinsic regional differences in light resetting within the SCN neural network, with the magnitude of acute phase shifts that underlie ongoing waveform changes being greatest in the ventrolateral SCN core, while the magnitude of subsequent period after-effects is greatest in the dorsomedial SCN shell.

A key question addressed in our study was how exactly light input changes the clock gene rhythms in the SCN to drive phase shifts and to achieve entrainment to external light cycles. Previous studies (*Messager et al., 2000*; *Schwartz et al., 2011*) have assayed the clock gene rhythms under lighting conditions by time-point sampling clock gene expression from populations of animals for each time point at low temporal resolution (e.g. 4 hr sampling intervals), making it difficult to fully capture dynamic waveform changes in the rhythms induced by light input. Using bioluminescent recording of PER2 expression at an interval of minutes, while optogenetically providing the SCN explant with simulated light signals, we directly showed that discrete light input induces acute induction of PER2 expression and differential waveform changes in PER2 rhythms in a phase-dependent manner, leading to a time-dependent phase shifts: high PER2 induction by early-night light exposure induces phase delays by elongating the falling phase of the molecular clockworks, while late-night light induces phase advances by accelerating the rising phase of the clockworks, and mid-day light does not affect the rhythm phase due to low PER2 induction. These differential waveform changes in specific PER2 phases differentiate PER2 waveforms under light cycles from those in free-running conditions. PER2 rhythms express typical sinusoidal waveforms in free-running conditions, whereas under light entrainment they show asymmetric waveforms with elongated falling phases, abbreviated rising phases and dramatic waveform changes aligned with light stimulation events. Given that a pulse of retinal light input can induce PER gene expression in the SCN and PER induction is required for the SCN clock resetting (*Shigeyoshi et al., 1997*; *Tischkau et al., 2003*), our results suggest that the SCN clock rhythms such as neural activity rhythms can phase-shift to acute light exposure and entrain to a light cycle via single and regular events of the aforementioned waveform changes, respectively, as a result of light induction of PER gene expression.

As daily phase shifts underlie adjustment of the endogenous period of circadian behavior to different day-night cycle lengths, daily waveform acceleration in the rising phase of PER2 aligned with dawn, and daily elongation in the falling phase of PER2 abundance aligned with dusk, express the entrainment of circadian PER2 rhythm to light cycle periods. Stable alignment of the SCN clock with repeated light stimulations at a phase-delaying circadian time achieved circadian entrainment to a longer-than-24 hr cycle mainly via repeated lengthening of the falling phase of PER2, whereas repeated shortening of the rising phase at a stable phase angle accompanied short T-cycle entrainment. For photoperiodic encoding, as brief light pulses defining dawn and dusk can entrain circadian

behavior, daily brief optogenetic simulations of light transitions at dawn and dusk shortened PER2 rising phase and lengthened the falling phase in the SCN, respectively, thereby matching PER2 rhythms to a 24 hr period. PER2 waveform widths were different between equinox and long skeleton photoperiods. Previous studies suggested that long photoperiod induces *Per* waveform broadening as high *Per* gene expression gets extended throughout long light periods (*Messager et al., 1999*; *Schaap et al., 2003*). However, our results from skeleton photoperiods suggest that light-dark transitions at dawn and dusk are sufficient to alter PER2 waveform width. Light exposures at earlier dawns and later dusks for long photoperiods drive more PER2 induction than do light exposures in equinox photoperiods, thus expanding PER2 waveform. This is consistent with waveform broadening of neural activity rhythms in animals under a long photoperiod (*VanderLeest et al., 2007*).

Another important point of this study is that plasticity in circadian behavior following alteration of lighting conditions is intrinsic to the SCN clockworks. Key principles of circadian entrainment and plasticity in mammals have been discovered by classical behavioral studies that assessed circadian locomotor behavior under different lighting conditions. Circadian behavior, however, is a product of multiple oscillators involving different brain areas, and the SCN receives both extensive feed-forward and feedback from other brain nuclei in situ (*Dibner et al., 2010*). Our results from ex vivo entrainment of isolated SCN strongly suggest that many of the classically studied forms of circadian plasticity and properties of entrainment apparently reside intrinsically within the SCN molecular and neural network themselves. As previously reported (*Jones et al., 2015*), differential phase responses of circadian rhythms depending on the timing of light input is evident even though the SCN neurons are directly stimulated, rather than them receiving retinal input. Furthermore, as entrainment of circadian behavior to different T-cycles is achieved by different phase angles of entrainment, the SCN clock itself establishes different entrained phase angles to the T-cycles, consistent with non-parametric model of entrainment (*Pittendrigh and Daan, 1976b*). Daily light stimulations fall in the phase-advancing zone (late falling PER2 phase) and the phase-delaying zone (late rising PER2 phase) during short and long T cycles, respectively. Also, the SCN clock itself can indeed be entrained directly by the light-dark transitions in skeleton photoperiods as mimicked by optogenetic stimulation, and a higher order aspect of entrainment, such as the bias of the circadian system to resolve ambiguous skeleton light cycles in favor of short-day entrainment (*Pittendrigh and Daan, 1976b*), also resides in the SCN clockworks.

We also uncovered whether and how the SCN clock itself expresses light-induced after-effects on the endogenous clock period. Acute phase shifts produce after-effects on the endogenous period of the SCN clock itself, and interestingly, the magnitude of the period after-effects (up to several hours) is larger than that at the level of behavioral rhythms (less than an hour) (*Pittendrigh and Daan, 1976a*). This suggests that the after-effects on circadian behavior following single light pulses become diminished as light signals are transmitted downstream of the SCN, or that the plasticity of the in vivo SCN is constrained by interactions with extra-SCN oscillators or circuits. In the case of entrainment to non-24 hr periods (T-cycles), surprisingly, we did not detect significant period after-effects in the SCN clock, although T-cycle entrainment produces most significant period after-effects in circadian behavior (*Pittendrigh and Daan, 1976b*). It is an unexpected finding that repeated stimulation does not produce large period after-effects, while single light stimulation does. Period after-effects of circadian locomotor behavior are expressed after more than a month of exposure to different T-cycles (*Azzi et al., 2014*; *Schwartz et al., 2011*), suggesting that our one-week entrainment paradigm might not be sufficient to drive T-cycle period aftereffects. Notably, previous findings indicate that the clock period of SCN explanted from animals entrained to T-cycles does not represent circadian behavioral period of those animals (*Aton et al., 2004*; *Azzi et al., 2017*; *Ciarleglio et al., 2014*; *Molyneux et al., 2008*), suggesting the possibility that extra-SCN clocks are a critical influence on the period after-effects of T-cycles. Our results from direct entrainment of isolated SCN to T cycles also suggest that there may be additional non-SCN influences underlying the T cycle period after-effects at the behavioral level.

We also revealed how light exposure alters the network state of the SCN clock to induce phase shifts and period plasticity, and directly showed that the SCN itself has subregional heterogeneity in clock-resetting capacity. Previously, it was assumed that retinal light input is mainly received by the ventral SCN, creating differences in light-induced clock resetting between the ventral and dorsal regions due to differential input (*Yan et al., 2007*). For example, in the in vivo SCN light induction of *Per1* mRNA expression is more prominent to the ventral region, whereas *Per2* induction is widespread (*Yan*

and Silver, 2002). More recent studies using intrinsically photosensitive retinal ganglion cell (ipRGC) labeling (*Chen et al., 2011*; *Fernandez et al., 2016*), however, showed that retinal projections are widespread across the entire SCN and neuronal activation following light exposure was ubiquitous, suggesting the possibility of intrinsic differences in responsiveness between the SCN subregions. Using pan-neuronal optogenetic stimulation that was synchronous across the entire SCN, we found that the lateral or ventrolateral SCN exhibits large phase responses and small period responses, while the medial or dorsomedial SCN exhibits small phase responses but large period plasticity, revealing an intrinsic nature of the clock-resetting heterogeneity in the SCN. As the VIP and AVP neurons are respectively located in the ventral and dorsal SCN, this suggests that regionally differential phase and period responses in the SCN might be derived from intrinsic differences between the VIP and AVP neuronal clocks. The period response, inversely correlated with the phase response, could serve to help the SCN recover back to its baseline network phase state from decreased synchrony following phase shifts. Decreased phase synchrony among SCN neurons following light exposure has been observed in the case of VIP-induced phase shifts (*Hamnett et al., 2019*) and entrainment to long photoperiods or non-24hr light cycles (*Azzi et al., 2017*; *Evans et al., 2013*). Interestingly, stimulating VIP neurons reset the ensemble SCN phase, while stimulating VPAC2 neurons located in the dorsal SCN do not (*Patton et al., 2020*), suggesting regionally different phase-resetting capacity. Future studies will be needed to address whether and how the subregional heterogeneity in light responsiveness fulfills encoding of various lighting conditions in the SCN.

Technical limitations of our methods are twofold. We used~ P12 SCN slices to achieve long-term monitoring of real-time clock gene rhythms throughout optogenetic entrainment. Although SCN astrocytes become fully mature by P20-25 and they are recently identified as an important component for entrainment in vivo (*Brancaccio et al., 2019*; *Brancaccio et al., 2017*; *Tso et al., 2017*), SCN maturity reaches near-adult levels at P12 in many aspects including retinal innervation, clock gene rhythmicity, neuropeptide expression profile, and photic responses (*Bedont and Blackshaw, 2015*). Additionally, our current system is not readily applicable to studying real-time SCN rhythms under a full optogenetic light cycle (e.g. 12 hr of stimulation every day) as excitation light for optogenetics interrupts bioluminescence detection. Moving forward, overcoming such a limitation will further extend the usage to studying SCN-intrinsic mechanisms of entrainment. Interestingly, real-time clock gene rhythms in the in vivo SCN re-entrain to a new light cycle during experimental jet lag (*Mei et al., 2018*). Future waveform analyses of clock gene rhythms in such an in vivo setup would allow comparison with our observations in the isolated SCN.

Lastly, our study provides technical contributions to studies of SCN entrainment and plasticity and to the study of neural plasticity in general. Our system that employs long-term red optogenetic stimulation and photomultiplier tube-based bioluminescence recording from cultured neural tissues enables ex vivo entrainment of the isolated SCN neural network over intervals of days to weeks in a similar way to light entrainment in animals, and high-resolution assessment of plasticity in the SCN clock gene rhythms throughout entrainment. As the SCN explant is more accessible to genetic or pharmacological manipulations than the in vivo SCN, our method will provide a great opportunity to further study molecular mechanisms of the SCN entrainment. Our system can also be adapted for long-term optogenetic stimulation with other bioluminescent readouts, such as bioluminescence resonance energy transfer (BRET) $Ca^{2+}$ sensors (*Suzuki et al., 2016*; *Yang et al., 2016*), and thus it is potentially widely applicable to studying induction of long-term neural plasticity in different regions of the brain.

# Materials and methods

**Key resources table**

| Reagent type (species) or resource | Designation | Source or reference | Identifiers | Additional information |
|---|---|---|---|---|
| Strain, strain background (*M. musculus*) | PER2::LUC | *Yoo et al., 2004* | RRID: IMSR_JAX:006852 | |

*Continued on next page*

*Continued*

| Reagent type (species) or resource | Designation | Source or reference | Identifiers | Additional information |
|---|---|---|---|---|
| Transfected construct (*M. musculus*) | AAV1-Syn-ChrimsonR-tdT | *Klapoetke et al., 2014* | Addgene viral prep # 59171-AAV1 | |
| Software, algorithm | MATLAB | Mathworks | RRID:SCR_001622 | |
| Software, algorithm | ClockLab Analysis | Actimetrics | Matlab-based ClockLab Analysis ver. 2.72 | |
| Software, algorithm | LumiCycle | Actimetrics | | |
| Software, algorithm | Signal Generator | Mhinstek | | |
| Software, algorithm | Prism | GraphPad | RRID:SCR_002798 | |
| Software, algorithm | SPSS | IBM | RRID:SCR_002865 | |
| Software, algorithm | OptoLumicycle | This study | https://github.com/SuilKim/OptoLumicycle (*Kim, 2021b* copy archived at swh:1:rev:fffae11b1135c5775ee40ffa48bed05315f5282e) | Codes for an integrated system of luminometry and optogenetic stimulation |
| Other | LED driver | Thorlabs | LEDD1B | |
| Other | Fiber-coupled LEDs | Thorlabs | M470F3 (470 nm), M625F2 (625 nm) | |
| Other | Fiber collimation package | Thorlabs | F230SMA-B | |
| Other | Multi-mode fiber cable (Ø1500µm, 0.39NA) | Thorlabs | M93L | |
| Other | Relay switch | Sensata-Crydom | DC60S5-B | |
| Other | Multi-functional I/O device | National Instruments | USB-6001 | |

## Animals and housing

P11-14 heterozygous PER2::LUC knock-in mice (*Yoo et al., 2004*) were used for organotypic slice culture and subsequent procedures as the PER2::LUC knock-in allele can alter circadian functions such as free-running period and entrainment (*Ralph et al., 2021*). All animals were housed in a 12:12 LD cycle, and had food and water provided ad libitum. Both male and female mice were used in experiments. Experiments were performed in accordance with the Vanderbilt University Institutional Animal Care and Use Committee and National Institutes of Health guidelines.

## Organotypic slice culture and AAV viral transduction

Removed brains were mounted and cut into coronal slices (300 µm) on a vibrating blade microtome (Leica) in cold HBSS supplemented with 100 U/ml penicillin/streptomycin, 10 mM HEPES, and 4.5 mM sodium bicarbonate. The SCN slices were dissected out and transferred onto a PTFE membrane insert (Millipore) in 35 mm culture dishes with 1.2 ml of DMEM (D5030, Sigma) supplemented with 3.5 g/L D-glucose, 2 mM Glutamax (Gibco), 10 mM HEPES, 25 U/ml penicillin/streptomycin, 2% B-27 Plus (Gibco), and 0.1 mM D-Luciferin sodium salt (Tocris). The SCN slice position was adjusted to the center of the dish and 1.5 µl AAV (pAAV1-Syn-ChrimsonR-tdT, Addgene) (*Klapoetke et al., 2014*) was placed directly onto the SCN slice. The culture dishes were then sealed with an optically clear PCR plate film (Bio-Rad) and maintained in a non-humidified incubator at 36.8 °C for about 10 days. The opsin expression was checked after about 10 days of viral transduction by imaging tdT fluorescence.

## Bioluminescence recording and in situ optogenetic stimulation

After viral transduction, bioluminescence from firefly luciferase in each of PER2::LUC SCN slices was recorded in 6 min intervals by a 32-channel/4-photomultiplier tube luminometer LumiCycle (Actimetrics) in a non-humidified, light-tight incubator at 36.8 °C. Baseline rhythms were recorded for at least three days before optogenetic stimulation. For optogenetic stimulation, 625 nm LED light (10 Hz, 10 ms pulse width, 1.5 mW/mm$^2$) was delivered at the center (6 mm illumination radius) of a target culture dish by a fiber-coupled LED (M625F2, Thorlabs). The LED was located outside the incubator and coupled to a multimode fiber cable (Ø1500μm, 0.39NA) (M93L, Thorlabs) and a fiber collimation package (F230SMA-B, Thorlabs) tethered above samples. Light pulses were generated by an LED driver (LEDD1B, Thorlabs) and a signal generator (Mhinstek). For remotely turning on and off the photomultiplier tubes in the luminometer, a relay switch (Sensata-Crydom) was added in the electrical circuit of the luminometer and connected to a multi-functional I/O device (National Instruments). Custom-written code (OptoLumicycle, *Kim, 2021a*) in Matlab (Mathworks) was used to access luminescence data collection software (Actimetrics), the multi-functional I/O device, and a signal generator software (Mhinstek). The Matlab code loaded a spreadsheet having stimulation settings and time schedules, and executed a series of events during optogenetic stimulation: pause bioluminescence recording, target positioning, stimulation initiation, stimulation termination, and resumption of the recording. For a long light exposure test, 12 h blue or red light pulses (10 Hz, 10 ms pulse width, 1.2 mW/ mm$^2$) were illuminated onto PER2::LUC SCN slices by LEDs (M470F3 or M625F2, respectively, Thorlabs) coupled to a multimode fiber cable (Ø1500μm, 0.39NA) (M93L, Thorlabs). Blue light-illuminated samples were given a medium change 2 days after illumination.

## Bioluminescence recording data analysis and visualization

Raw bioluminescence data were baseline-subtracted using 24 hr running averages and smoothed by 2.4 hr moving averages using LumiCycle Analysis software (Actimetrics), and then they were loaded as normalized actograms into Matlab-run ClockLab software (Actimetrics) for further analyses. Phase shifts were determined as the time difference between the observed post-stimulation peak of the bioluminescence rhythm and the predicted peak from a linear regression of at least three cycles before stimulation. Period changes were determined as the difference in the period length between the pre-stim and the post-stim cycles. Period length was calculated using a linear regression of at least three peaks or using Lomb-Scargle periodogram in ClockLab software. If peaks were not obvious, the period length was determined using half-maxes of the rising phase. Acrophases were calculated using ClockLab software. For data visualization, smoothed and baseline-subtracted bioluminescence rhythms were represented as double-plotted actograms normalized to min and max values of the data for each line of the actograms, using Excel (Microsoft) and Prism (Graphpad). The actograms were 24h-plotted unless otherwise stated. Bioluminescence rhythms of 12 hr light-exposed slices were visualized using Excel and LumiCycle Analysis software. For quantifying the effect of a long light exposure on the rhythm amplitude, the amplitude of the first post-treatment cycle was normalized to the amplitude of the last pre-treatment cycle. For waveform analyses of single phase shifts, smoothed bioluminescence rhythms with and without stimulation were normalized to extrapolated peak and trough values based on the average dampening rate of the rhythm amplitude over time. Then the rhythms with stimulation were subtracted from those without stimulation. Normalized induction following stimulation was calculated as the first-cycle amplitude of the subtraction data. For waveform analyses of bioluminescence rhythms during T cycle entrainment, smoothed bioluminescence rhythms before T22/T25 entrainment and during T22 entrainment were normalized to peak and trough values, and the rhythms during T25 entrainment were normalized to actual troughs and interpolated peaks based on the first cycle of the free-running rhythms released in constant darkness. For waveform analyses of bioluminescence rhythms during skeleton photoperiods, the smoothed first derivatives were taken from the smoothed, baseline-subtracted bioluminescence rhythms normalized to min and max values, using Excel and Prism. The duration of the rising and falling phases was determined using the time duration between the zero crossings of the first derivatives. If the first derivatives were close to zero but did not make a zero crossing due to an increase immediately following an optogenetic stimulation, the local minimum was defined as the end of the rising phase. Waveform data were visualized using Excel and Prism.

## Time-lapse bioluminescence imaging with optogenetic stimulation and data analyses

SCN slices expressing PER2::LUC and pAAV-ChrimsonR-tdT were cultured in the sealed 35 mm dishes as described above, and the dishes were transferred into a temperature-controlled chamber of the LV200 microscope system (Olympus) equipped with an imaging software (CellSens, Olympus), an EM-CCD camera (Hamamatsu) and an LED light illumination system (CoolLED). Bioluminescent images were acquired every ten minutes using a 40 x objective lens and 0.2 x tube lens. For optogenetic stimulation, 10 Hz 635 nm LED light pulses with 10 ms pulse width were illuminated for 15 min using real-time controllers (Olympus) and the experimental manager of CellSens imaging software. Stimulation time was determined using at least three cycles of PER2::LUC bioluminescence rhythms before stimulation. For regional analyses in ImageJ (NIH), background noise in bioluminescent image series were removed using grouped Z projections with group size two and minimum intensity method, and 15 × 15 pixel ROIs were generated based on average intensity projections of the image series. Mean intensity data from each ROI was 0th-order smoothed in Prism, and peak times of the bioluminescence rhythms were determined using Excel. To create a circadian phase map, peak time data were normalized to the mean and plotted as a rainbow-colored heat map in Prism. A circadian period map was created using period values calculated from three cycle peaks for each ROI. Phase shifts and period changes were calculated as described above. To determine the overall phase dispersion within the SCN slice, circular variance (= 1 − mean vector length) was calculated and Rayleigh plots were generated using Oriana (Kovach Computing Services).

For cluster analyses in each dimension (e.g. phase shift), two clusters were formed from each SCN slice using Ward's minimum variance method in SPSS (IBM) to separate ROIs with larger values from ROIs with smaller values, and then two-clustered SCN slices were color-coded in Excel. The color-coded clusters were group-averaged using ImageJ, and the group-averaged was clustered again in SPSS to divide into three regions. To calculate the difference in each dimension among the three clusters, bioluminescent image series were re-analyzed using ROIs corresponding to each cluster.

## Statistical analysis

All statistical analyses were performed in Prism (Graphpad). Statistical tests used for each experiment are provided in the figure legend. Data are presented as a mean ± standard error of mean (SEM) and differences between groups were considered statistically significant when $p < 0.05$.

## Acknowledgements

We are grateful to D Sprinzen for helping integrate optogenetic stimulation apparatus with a multi-channel luminometric device, and to M Hastings for his help in improving our SCN culture medium and choosing parts for the LV200 system. This study was supported by National Institute of Health grant R01 GM117650 to DGM; Vanderbilt International Scholarship to SK.

## Additional information

### Funding

| Funder | Grant reference number | Author |
| --- | --- | --- |
| National Institute of General Medical Sciences | R01GM117650 | Douglas G McMahon |
| Vanderbilt University | Vanderbilt International Scholarship | Suil Kim |

The funders had no role in study design, data collection and interpretation, or the decision to submit the work for publication.

### Author contributions

Suil Kim, Conceptualization, Formal analysis, Investigation, Methodology, Project administration, Visualization, Writing - original draft, Writing - review and editing; Douglas G McMahon, Conceptualization,

Funding acquisition, Project administration, Resources, Supervision, Writing - original draft, Writing - review and editing

### Author ORCIDs
Suil Kim http://orcid.org/0000-0002-2739-1238
Douglas G McMahon http://orcid.org/0000-0002-0645-2281

### Ethics

This study was performed in strict accordance with the recommendations in the Guide for the Care and Use of Laboratory Animals of the National Institutes of Health. All of the animals were handled according to approved institutional animal care and use committee (IACUC) protocols (M1800065-01) of Vanderbilt University.

### Decision letter and Author response
Decision letter https://doi.org/10.7554/eLife.70137.sa1
Author response https://doi.org/10.7554/eLife.70137.sa2

---

## Additional files

### Supplementary files
• Transparent reporting form

### Data availability
All data generated or analyzed during this study are included in the manuscript and supporting files. Source data files have been provided for Figures 1 to 6.

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
