## [Editor Report]

The work by Kim and McMahon describes a useful ex vivo approach to address fundamental circadian mechanisms. Their strategy allows optogenetic manipulation of suprachiasmatic nucleus (SCN) slices with red-light, in order to mimic light input. By conducting classic entrainment protocols they are able to correlate data obtained in whole animals, with ex-vivo changes in core-clock gene expression (PER2LUC). Thus, they assess how some key organismal circadian properties, related to light entrainment, can be recapitulated ex-vivo in isolated SCN slices, while other ones may depend on more complex organismal physiological intricate networks. The major conclusions of the work include: rapid optogenetic activation of SCN neurons results in changes in the SCN PER2 waveform that correlate with changes in behavior induced by seasonal or non-24h environmental cues.

---

## [Decision Letter]

**Decision letter after peer review:**

Thank you for submitting your article "Light sets the brain's daily clock by quickening and slowing regional nodes of the molecular clockworks at dawn and dusk" for consideration by *eLife*. Your article has been reviewed by 3 peer reviewers, one of whom is a member of our Board of Reviewing Editors, and the evaluation has been overseen by Ronald Calabrese as the Senior Editor. The following individual involved in review of your submission has agreed to reveal their identity: Erik D Herzog (Reviewer #2).

Essential revisions:

1) It is important to provide a more comprehensive analysis and discussion of the data considering prior papers that have utilized similar approaches (i.e optogenetic perturbation affecting SCN firing and clock dynamics), or that have pursued similar questions related to light entrainment. I.e. there have been interesting publications focusing on SCN explants coming from animals kept under diverse photoperiods, or in vivo monitoring (in freely moving animals) of the SCN (via clock reporters or electrophysiological recordings) of animals subjected to different light stimulation protocols. The authors should also consider whether there are features that differ between the behavior and the isolated SCN (i.e. did the rate of entrainment in vivo differ from in vitro?).

2) It is strongly suggested that the authors use the same phase markers in all experiments and at least one additional phase marker, preferably 'center of gravity' which effectively incorporates changes in waveform shape. Suggestions also include measuring period with an additional method (particularly as only a few days are measured for the analyses), as well as considering other interpretations for some of their observations (how spatial expression of ChR may explain the differences in dSCN and vSCN PER2 patterns following optostimulation). Importantly, to rule out the latter they should quantify the fraction of SCN cells that express their optogenetic driver.

3) Modify several aspects of the text (avoid excessive clock-terms/jargon), clarify some figures (i.e. Figure 4), and explain the rationale behind the choice of T-cycles and photocycles that were tested, to help a general audience to better understand key aspects of the work.

4) The authors should seek better evidence that optogenetic stimulation did not kill cells, impair redox state, or PER2 cycling.

5) The authors should address the concern that phase shifts were calculated immediately after perturbation and compare their findings to any steady-state phase shifts after correcting for any induced period changes (see Rev #2, point 4).

*Reviewer #1:*

The work by Kim and McMahon describes an exciting combination of methods that allow obtaining accurate circadian data from suprachiasmatic nucleus (SCN) slices over many days, while exploring how key clock parameters are affected by perturbations mimicking regular light cues, ex vivo. In order to do so, the authors adopt an optogenetic strategy based on ChrimsonR: a light-gated cation channel that upon red-light activate neurons. The choice of red-light over blue was not casual, as the latter can have negative effects on the primary culture if prolonged stimulation is used. Previous studies that have focused on the effect of different photoperiods or light-perturbations on core-clock dynamics in the SCN were either indirect measurements (running-wheel activity), had limited resolutions (i.e. SCN samples collected from different animals every 4 h), or implied monitoring real-time reporters in SCN slices, after the particular entrainment or perturbation had occurred in the animals. With the new approach presented by Kim and McMahon it is possible to monitor any reporter of choice (in this case PER2LUC), and see what happens in response to particular light perturbations (by mimicking neural firing through the optogenetic system). Thus, the authors are now able to obtain accurate information on PER2LUC dynamics upon defined perturbations, that they dissect focusing on changes in oscillations waveforms: they analyze what happens in terms of phase and period when discrete perturbations (designed to mimic a 15 min light pulse seen by the animal) are given at different times throughout 24 h (reproducing cycles of different T). The analyses also included exploring the effect of skeleton cycles of different photoperiods. Notably, they can reproduce with this ex-vivo perturbations several circadian aspects described in foundational papers (i.e. Pittendrigh and Daan, 1976), while also observing phenomena that are different from what one may expect, reflecting more complex organismal responses. Thus, among some of the interesting findings is that the ability (or bias) to resolve ambiguous skeleton light cycles favoring short-day entrainment, appears to reside in the SCN clockworks, as deduced from their ex-vivo work. Also, their observations indicate that repeated stimulation does not produce large period after-effects, whereas single light stimulation does so. Finally, their approach also allowed them we observe different responses in lateral or ventrolateral vs medial or dorsomedial SCN changes in phase or period responses revealing an intrinsic nature of the clock-resetting heterogeneity in the SCN.

While the work is interesting, it is key to enrich the analysis and discussion of the data under the light of other existing publications that have utilized similar approaches (i.e optogenetic perturbation affecting SCN firing and clock dynamics), or that have pursued addressing similar questions regarding light entrainment. Thus, the latter includes distinct publications analyzing SCN slices coming from animals kept under diverse photoperiods, or in vivo monitoring (in freely moving animals) of the SCN (via clock reporters or electrophysiological recorders) of animals subjected to different light stimulation protocols..

In addition, it becomes important to better explain for a general audience the choice of how T-cycles and photocycles were established i.e. with single light pulses or skeleton photoperiod as opposed to classic LD cycles. Indeed, one of the advantages of implementing this red-light optogenetic system is that prolonged light stimulation can be applied.

Therefore it is puzzling to see that the authors only used it for discrete 15 min light pulses as opposed to light (red-light) mimicking full photoperiods and not just skeleton ones.

1 – "We found that PER2::LUC bioluminescence became arrhythmic following the prolonged blue light exposure (Figure 1A, B). This effect was not reversible with a medium change (Figure 1A), suggesting that long-term blue light exposure per se can elicit photodynamic damage of SCN slice cultures".

The authors utilized (470nm, 10Hz frequency, 10ms width, 1.2mW/mm2) for 12h. Could they comment whether (i) lower intensity (or change in the duty cycle) could be used (to obtain similar response if they were containing an optogenetic construct) (2) Better define and elaborate what the "photodynamic damage" may be. i.e is it possible to use any marker that would be informative of apoptosis, mitochondrial damage, DNA fragmentation etc?

2 – The authors ought to discuss their results in the context of other relevant work that has used similar approaches or pursued similar questions: i.e. Patton et al., PMID: 32636383, utilizing ChR to control with blue-light firing of VIP and VPAC2 neurons; or Mei et al., PMID: 29610316 following Per2 and Cry1 in vivo dynamics (by monitoring reporters in freely moving animals) under some particular light conditions; or the work of VanderLeest et al., PMID: 17320387 conducting in vivo electrophysiological recordings in animals under short or long days., among other papers that may be addressing similar questions.

3 – Figure 1: ChrimsonR-tdT is mentioned in the figure legend, yet, it is not properly explained in the manuscript, although the ChrimsonR is at the heart of the optogenetic system utilized in this work.

4 – Supplement 1. It is nice to evidence the major effect that stabilized glutamine, and utilizing slices from younger mice have on rhythms. There are a couple of things that the authors could further comment on this:

a) There seems to be a slight phase delay in the P12 -Ala-Gln traces. Could the authors comment on this?

b) The great improvement in signal robustness and amplitude results from changing two variables at once (age of the animals source of SCNs and media composition). Therefore, it is hard to weight on the relative contribution of each variable to rhythm quality. Therefore, if the authors have data allowing to understand which variable is more critical it would be great to include that in the manuscript. Importantly, other groups doing SCN explants analyses also use young animals (i.e P8-P10 PMID: 32636383; P3 PMID: 25186748), or already use glutamax (see below). Therefore, what would be the added value/novelty of the presented data (Figure 1, suppl. 1)?

c) The authors use glutamax 0.2 mM, whereas other experiments (i.e. Patton et al., PMID: 32636383) use 2 mM glutamax, also achieving good readings for well over 15 days.

5 – Figure 1—figure supplement 2. The system that the authors have elaborated to optogenetically control their samples while also monitoring PER2-LUC expression is extremely useful and for sure will be a valuable resource for other colleagues in the field. Therefore, it is recommended that they provide more technical details of the setup (actual photos of the setup, radius of the illuminated region etc), and make available the Matlab code used to run the hardware/retrieve the luc data.

6 – Figure 2A. It appears that the traces corresponding to the CT21 intervention are showing an altered period from even before the stimulation. Could the authors check and comment on that?

7 – Page 8: "Further, the acute phase shifts were accompanied by a subsequent sustained change in the free-running circadian period". From the data it is not fully clear:

1) When (after starting the SCN cultures) was red-light stimulation applied (from Figure 2 it seems that on the 3rd day of recording). Please explicitly mention it in the main text.

2) How was the period calculated (based on how many days of recording after stimulations; the authors only mention "several days"). This is important, to make sure that the calculated changes in period do not correspond to, nor are affected by, transients. This becomes quite relevant as subsequent experiments involving repeated perturbations do not show period effects.

8 – Page 8: "Stimulation at CT14 elongated the falling phase to induce a phase delay"

Another way to describe what is seen is that stimulation causes a new rise, delaying the falling phase and therefore causing a phase delay.

9 – Page 8: "whereas stimulation at CT21 prematurely ended the falling phase and accelerated the rising phase to induce a phase advance".

Looking at the data it makes it harder to see such a "premature end" of the falling phase, as the stimulus is given around the time of the trough. I agree that it accelerates de rising though, leading to a phase advance.

10 – Page 8: Together, differential waveform changes in PER2 rhythms induce time-dependent phase responses to light stimulation in the SCN".

The way this phrase is constructed seems to implicitly denote causation on the waveform properties of Per2, where actually a more accurate description would be related to the changes on the state variable itself.

11 – Figure 3E. The traces corresponding to the "before" T22 or T25 cycles (black lines) appear to be different in both experimental sets. Thus, while in the left graph the pre-T curve has a peak centered around 18, in the right graph the peak is centered around 15 or so.

12 – It is not clear why the authors chose for the entrainment protocol of different T only 1 light pulse, instead of opting for a skeleton photoperiod (although they did use skeleton photoperiod in following experiments). Moreover, based on the improved properties of this red-light optogenetic systems the authors could have even tried subjecting the SCN slices to full light:dark cycles of different T (i.e 12 h light, 12 dark), as one of the points of utilizing a red-light optogenetic system, was to actually be able to activate the system for many hours (with no cytotoxicity). Could the authors comment on the logic behind their choice and the limitations of only exploring T cycles pushed with one light pulse (as opposed to skeleton or full photoperiods) ?. Likewise, their system, as opposed to the ones relaying on blue-light, allows creating more realistic LD regimes (i.e full 12 h of light, as opposed to a skeleton one).

13 – Page 18: "Group averaged clustering analysis identified a lateral-medial axis for phase shifts and a ventrolateral-dorsomedial axis for period changes. In terms of changes in phase coupling, the phase variance within the SCN was increased following stimulation (Figure 6B), indicating a larger phase desynchrony in the SCN."

This is an interesting inference from the data. Could the authors comment more on how this compares with prior SCN-luc section studies derived from animals kept under different light regimes?

14 – Page 20: " we directly showed that discrete light input induces acute induction of PER2 expression and differential waveform changes in PER2 rhythms in a phase-dependent manner, leading to a phase-dependent resetting response.."

Instead of talking about resetting, wouldn't it be more appropriate to talk about phase shifts (at least for the intensity of the utilized light-pulses)?

15 – Materials and methods:

– Could the authors comment on why they used heterozygous (as opposed to homozygous) PER2::LUC mice? Were both male and female individuals used for SCN preparations?

– "Custom-written code in Matlab (Mathworks) was used to access luminescence data collection software (Actimetrics), the multifunctional I/O device, and a signal generator software (Mhinstek)." Available upon request?*Reviewer #2:*

The authors introduce a new system to record clock gene expression (as luminescence) for weeks while using optogenetic stimulation to test whether the isolated suprachiasmatic nucleus (SCN) exhibits circadian properties of period aftereffects, entrainment to non-24 h cycles and to skeleton light cycles previously described in vivo. They present the results from well-designed experiments which validate the technique and reveal that these properties are indeed intrinsic to the SCN slice. It is nice to see classical chronobiology concepts revisited and attributed to changes in gene expression of a small neural network. The major conclusions include: rapid optogenetic activation of SCN neurons results in changes in SCN PER2 waveform that correlate with changes in behavior induced by seasonal or non-24h environmental cues. This manuscript provides exciting results with excellent experimental design. The relevance of rapid changes in firing to rapid changes in PER2 and, ultimately, long-term changes in phase and period could be made more clear for a general readership. The manuscript will be improved when the authors address some major concerns with the analysis.

1) The authors conclude that each feature of circadian behavior that they sought to attribute to the SCN was, indeed, found in the isolated SCN. The authors should consider whether there are features that differ between the behavior and the isolated SCN. For example, did the rate of entrainment in vivo differ from in vitro? Importantly, to reach their conclusions, they use different phase markers – peak and mid-point of rising phases- in different places in the manuscript. Was this necessary to reach their conclusions?

2) To measure changes in period and phase, the authors should justify their choice for phase marker and why they change this for some experiments. In the context of this manuscript, the change in waveform shape can lead to over-estimation of phase and period differences [cf., Daan S, Oklejewicz M. The precision of circadian clocks: assessment and analysis in Syrian hamsters. Chronobiol Int. 2003]. We suggest that the authors use the same phase markers in all experiments and at least one additional phase marker, preferably 'center of gravity' which effectively incorporates changes in waveform shape. Observing similar results with 'center of gravity' gives a better picture of the magnitude of effects in response to optogenetic stimulation that the authors report here. [Kenagy, G. J. 1980. Center-of-gravity of circadian activity and its relation to free-running period in two rodent species. J. Interdiscip. Cycle Res.]. Because the authors use only 3 days of data to estimate the circadian period, they should use at least one additional method to measure the period, preferably a method like wavelet, periodogram or FFT which has different assumptions. [cf., Maria J. Costa, Bärbel Finkenstädt, Véronique Roche, Francis Lévi, Peter D. Gould, Julia Foreman, Karen Halliday, Anthony Hall, David A. Rand, Inference on periodicity of circadian time series, Biostatistics, Volume 14, Issue 4, September 2013, Pages 792-806. Also: Zielinski T, Moore AM, Troup E, Halliday KJ, Millar AJ (2014) Strengths and Limitations of Period Estimation Methods for Circadian Data. PLOS ONE 9(5): e96462.]

3) Figure 1 can be moved to supplementary results and, in fact, raises some concerns. The major point of this methodological figure is that the treatment (red optogenetic stimulation) is non-toxic, but the results are not convincing. With only three replicates, it appears the amplitude was more variable and trending to reduced following red light stimulation. The authors should seek better evidence that optogenetic stimulation did not kill cells, impair redox state, or PER2 cycling.

4) Figure 2 validates what has been shown by this lab and others: Optogenetic stimulation of the SCN in vitro or in vivo can shift circadian rhythms. The authors should address the concern that phase shifts were calculated immediately after perturbation and compare their findings to any steady-state phase shifts after correcting for any induced period changes. This includes changes to the measurements in Figures 3, 4 and 6. This will allow the authors to: 1) measure similarities between in vitro and in vivo phase shifting and 2) quantify how much of the shift is due to a period change vs. an instantaneous phase shift. For example, the authors report large magnitude (4-5h) phase-shifts following stimulation at CT14 and CT21 in Figure 1. However, in Figure 6A most ROIs indicate a phase shift of only about 2h and very few ~3h. Similarly, in 6B, only a small proportion of ROIs indicate 4+h phase shift. Is this the result of differences in how the data were collected or analyzed?

5) Can the authors relate the spatial (throughout the SCN) and temporal (10Hz, 10ms pulses) optogenetic stimulation to photic activation of the SCN? To begin to do this, the authors should quantify the fraction of SCN cells that express their optogenetic driver. They should make it clear in the Results that they depend on viral transduction and the synapsin promoter (appears on page 22 of the manuscript while the tool is used starting on page 3). Presumably this results in neuron-specific expression of their ChrimsonR. For example, does the spatial expression of ChR explain the differences in dSCN and vSCN PER2 patterns following optostimulation (Figure 6)?

6) For fold change data presented in Figures 2-4, comparison of differences between rising and falling durations becomes complicated with increasing light conditions. Since the authors are focusing on the change in symmetry of the waveforms, the authors can reanalyse the data as a ratio of the two durations – 'rising phase duration/falling phase duration' before and after stimulation instead of analysing each of the phases' duration separately.

8) Page 7: "Optogenetic stimulation…. advances of about 4hr, respectively (Figure 2C)" There is considerable variation (upto 2.5 h) in phase shifts between animals. Instead of presenting the data as 'no phase shift' and 'about 4 h', please present the data as mean + sem here and in other places as well.

*Reviewer #3:*

In this study, the authors used long-term organotypic explant culture, cyclic red light optogenetic stimulation, and the PER2 bioluminescent reporter, to assess how the clock gene rhythms in the ex vivo SCN change in real time to achieve entrainment to light cycles. The use of the red wavelengths to drive optogenetic stimulation is an important advance as the blue light illumination common used produces phototoxicity in culture. The great strength of this work is that the authors provide data which enables us to visualize how the SCN clock responds to some of the classic environmental manipulations. To some readers, the weakness would be that the authors did not use their preparation to look at mechanistic questions. Still, in balance, the data showing how the hands of the clock are actually moving in response to classic entrainment protocols used for behavioural analysis is a major advance.

Figure 1 shows the methodology used and provides compelling data on the benefits of the red light stimulation. Maybe this data could be placed in supplemental information as it is really about the methodology. Still it does nicely illustrate the costs of the commonly used blue light stimulation.

In Figure 2, the show that the optogenetic stimulation alters the waveform of SCN PER2 rhythm following the classic phase response curve. With single 15min 10Hz optogenetic stimulation of SCN slices producing phase delays (CT 16), advances (CT21), and no phase shifts (CT 6). This data represents an important control and demonstrates that their system mimics the effects of light.

To me, Figure 3 is the beginning off novel experiments. The authors used their optogenetic system to entrain PER2::LUC Rhythms in the SCN to different cycle lengths (22 and 25 hrs). I believe that this is a novel set of experiments and shows how the SCN waveform is altered to synchronize to different T-cycles.

There is a long history of using skeleton photoperiods to mimic short (8 hrs per day), equinox (LD 12:12), or long (16 hrs per day). The authors used their optogenetic system to entrain PER2::LUC Rhythms to these different photoperiods. Entrainment to different skeleton photoperiods altered the molecular waveform of the SCN clock (Figure 4). Stimulations at dawn shortened the PER2::LUC rising phase, while stimulations at dusk lengthened the falling phase and set the phase angle of entrainment. Of course, these findings are a beautiful visualization of conceptual principles laid out by Pittendrigh and Daan. With Figure 4, there needs to be a little clarification to the graphs. We know that the authors are using 2 red light treatments to mimic dawn and dusk. So why three red lines? This was confuse when I was looking at the figure.

It is well appreciated that the SCN is made of a number of cell types and usefully divided at the network level by VIP+ and AVP+ cell populations. The authors then assessed how the circadian phase shifts by single light pulses impact the network state of the SCN clock. They combined optogenetic stimulation of the SCN with spatially imaging real-time PER2::LUC bioluminescence in using a microscope to provide regional information. Following CT14 stimulation, the authors report that the lateral region showed larger phase delays than did the medial region. In contrast, period lengthening effects were more prominent in the medial SCN than in the lateral SCN. Following CT21 stimulation, the authors find the ventral SCN showed larger phase advances and smaller period shortening than did the dorsomedial or dorsal SCN. So that phase shifting data indicate an inverse correlation in magnitude between the phase shift and the period change among SCN subregions. Cluster analysis also identified a ventrolateral-dorsomedial axis for phase shifts and a ventral-dorsal axis for period changes. I believe that the reader could have used little more help with understanding the analysis of these experiments.

Overall, the use of red optogenetic stimulation provides a technical advance over the blue wavelengths that are commonly used due to toxicity.

The authors make use of this technology to illustrate beautifully how the SCN clock response to phase shifting stimuli, T- cycles, and skeleton photoperiods. Even a phenomenon as esoteric as phase jumps is reproduced in their cultures. The authors go on to demonstrate regional differences in how the SCN cell populations respond to phase shifts.

Many of the findings are predicted by prior conceptual work in the field. But these prediction had not been put to the test so clearly before. To me this is a strength of the work.

The writing and figures are very clear. Some jargon needs to be cleaned up. For example, "transforms the waveform…to highly asymmetrical trajectories" or "regional nodes" sound more suited to a modelling study than empirical work.

As I mentioned above, there may be too many red lines in some of their figures! The analysis of the data shown in Figure 6 needs a little more explanation for a general audience.

The authors should at least discuss the possible impact of age. As I understand it, these cultures are P12 which is early in development.

Also, is the sex of the animals mentioned?

[Editors’ note: further revisions were suggested prior to acceptance, as described below.]

Thank you for resubmitting your article "Light sets the brain's daily clock by regional quickening and slowing the molecular clockworks at dawn and dusk" for consideration by *eLife*. Your article has been reviewed by 3 peer reviewers, including Luis F Larrondo as Reviewing Editor and Reviewer #1, and the evaluation has been overseen by Ronald Calabrese as the Senior Editor. The following individuals involved in review of your submission have agreed to reveal their identity: Erik D Herzog (Reviewer #2); Christopher Colwell (Reviewer #3).

Essential revisions:

Thanks a lot, Dr. Kim and Dr. McMahon, for this revised version that has successfully addressed most of the points that were previously raised. All three reviewers appreciated the modifications in the text that have helped clarify key aspects of the work and made it more accessible to a broader audience. As suggested by Rev#2, there are two aspects that still need further clarification /editing.

1) It is recommended to reword parts of the discussion relative to the comparison of in vivo ex vivo data, particularly when it comes to establishing possible causality. Whereas, the presented dataset is beautiful and establishes a nice correlation, yet there may be more intricate mechanisms in in vivo conditions.

2) The manuscript still concludes that the ChrimonR-based method is less toxic for in vitro optogenetic perturbation of the SCN than others based on blue light. As indicated by Rev#2 this may be too strong of a conclusion, as "toxicity" was only indirectly inferred from luciferase arrhythmic signals and assessed only after prolonged light-stimulation (which is not utilized in skeleton photoperiods). While this could be addressed by additional experiments that would actually establish such cytotoxicity, it would be also possible to rephrase parts of this topic.

Indeed, the data presented in the manuscript strongly suggests that prolonged stimulation with blue light is not an option for conducting full photoperiod circadian optogenetic studies. Yet, the dataset does not provide strong proof on whether this is due to cytotoxicity or another reason. Nevertheless, as the blue-light protocol tested in the manuscript (12 h pulse) is not directly comparable with the ones used for red-light throughout the paper (15 min pulses) this suggests that -potentially- one could have used a blue-light system and yet get all experiments to work just fine. But a valid, and relevant point is that Figure 1 shows that in a long optogenetic stimulation (12 h) the red-light systems outruns the blue-light one (albeit such stimulation protocol is not used later on throughout the paper, as the technical system does not allow one to stimulate and record at the same time). Thus, the data seem to strongly suggest that the ChrimsonR system has the "potential" of being more versatile as the quality of the rhythms are not affected by prolonged red-light stimulation, and would allow to test and compare (in future experiments) full photocycles (8:16, 12:12, 16:8) and skeleton ones, something that would be extremely difficult with blue-light.

*Reviewer #1:*

The manuscript has included important changes that help grasp the main findings of the work. Thus, it describes an ex vivo optogenetic experimental platform that allows precise optogenetic stimulation of explanted SCN slices while tracking clock gene expression with high temporal resolution. Importantly, the use of red-light does not bear some of the problems of cytotoxicity associated with blue-light and therefore allows multiple cycles of optogenetic modulation.

The most critical points raised by reviewers were successfully addressed.

*Reviewer #2:*

The authors are commended for their revisions to the text and analyses to accommodate comments from the editor and reviewers. They retain the same conclusions: the SCN in vitro entrains with many of the same features of circadian systems in vivo to different photoperiods. I recommend they reconsider their decisions on two major points they feel are important to the paper:

1. The authors now include additional similarities between SCN entrainment in vitro to optogenetic activation pulses and in vivo entrainment to long photoperiods. They find parallels between skeleton photoperiods in vivo and in vitro (e.g. psi jump) and period after effects to single pulses in vivo and in vitro. These correlated changes in PER2 expression in the cultured SCN should not, however, be used as evidence for their sufficiency or necessity in photic entrainment in vivo. I recommend removing causal arguments such as. "Daily waveform changes are sufficient to entrain to simulated winter and summer photoperiods, and to non-24h periods. SCN imaging further reveals that acute waveform shifts are greatest in the ventrolateral SCN, while period effects are greatest in the dorsomedial SCN." (abstract) and "PER2 rhythms in the SCN entrain to external light cycles via contraction of the rising phase and elongation of the falling phase depending on the timing of light exposure, and show how canonical features of light-induced plasticity in circadian behavior are expressed at the clock gene level. Circadian behavioral plasticity and regional heterogeneity of light responsiveness are intrinsic to the SCN clockworks." (Introduction).

2. They conclude they have developed a method that is less toxic for in vitro optogenetic perturbation of the SCN. Two reviewers challenged the authors to be more quantitative and thorough before concluding that the use of ChrimsonR and their modified culture media offer improvements over prior methods. The authors dedicate Figure 1 and Supplementary Figures 1 and 2 to this point. Figure 1 clearly shows that 12h of blue light dramatically decreased the bioluminescent reporter and a medium change did not rescue expression in 1-3 examples. Suppl Figure 1 provides a single comparison of young SCN + modified media to an adult SCN without modified. The authors use strong terms like "phototoxicity" and "photodynamic damage" and "build-up of toxic ammonia" without showing any damage to the SCN beyond loss of bioluminescence following 12 h exposure to blue light (a treatment they do not use for any of the subsequent experiments in the paper). They may be right that 12 h of blue light (10 Hz, 10 ms flashes) is incompatible with studies that aim to activate the SCN in vitro. However, we need more information about the intensity of the blue light used and whether dimmer light could be used to activate ChR with results similar to ChrimsonR. To argue ChrimsonR offers advantages over ChR, the paper would benefit from more than three replicates of the 12 h effect and, most relevant to this paper, evidence that the authors cannot measure the changes in PER2::Luc with skeleton photoperiods applied to ChR. As presented, the authors are welcome to justify their choice of media and CrimsonR, but they should not encourage researchers to change their media and optogenetic probe to avoid toxic side-effects.

*Reviewer #3:*

In my view, the authors have done an excellent job responding to the reviewer's concerns and the revised manuscript is ready for publication.

The great strength of this work is that the authors provide data which enables us to visualize how the SCN clock responds to some of the classic environmental manipulations. To some readers, the weakness would be that the authors did not use their preparation to look at mechanistic questions. Still, in balance, the data showing how the hands of the clock are actually moving in response to classic entrainment protocols used for behavioral analysis is a major advance.

I look forward to going through this study with the students in my laboratory.

---

## [Author Response]

Essential revisions:1) It is important to provide a more comprehensive analysis and discussion of the data considering prior papers that have utilized similar approaches (i.e optogenetic perturbation affecting SCN firing and clock dynamics), or that have pursued similar questions related to light entrainment. I.e. there have been interesting publications focusing on SCN explants coming from animals kept under diverse photoperiods, or in vivo monitoring (in freely moving animals) of the SCN (via clock reporters or electrophysiological recordings) of animals subjected to different light stimulation protocols. The authors should also consider whether there are features that differ between the behavior and the isolated SCN (i.e. did the rate of entrainment in vivo differ from in vitro?).

We have revised the text to better discuss our results in the context of previous relevant research findings adding the following passages (indicate section, page, and lines for each):

“Light exposures at earlier dawns and later dusks for long photoperiods drive more PER2 induction than do light exposures in equinox photoperiods, thus expanding PER2 waveform. This is consistent with waveform broadening of neural activity rhythms in animals under a long photoperiod (VanderLeest et al., 2007).” (Discussion, p. 11, lines 438-439)

“Decreased phase synchrony among SCN neurons following light exposure has been observed in the case of VIP-induced phase shifts (Hamnett et al., 2019) and entrainment to long photoperiods or non-24h light cycles (Azzi et al., 2017; Evans et al., 2013). Interestingly, stimulating VIP neurons reset the ensemble SCN phase, while stimulating VPAC2 neurons located in the dorsal SCN do not (Patton et al., 2020), suggesting regionally different phase-resetting capacity.” (Discussion, p. 12, lines 501-506)

“Interestingly, real-time clock gene rhythms in the in vivo SCN re-entrain to a new light cycle during experimental jet lag (Mei et al., 2018). Future waveform analyses of clock gene rhythms in such an in vivo setup would allow comparison with our observations in the isolated SCN.” (Discussion, p. 13, lines 518-521)

The ex vivo SCN rhythms reflect many canonical features of circadian rhythms in vivo such as time-dependent phase responses and entrainment properties, as shown in our work. However, the ex vivo SCN slice preparation receives direct neuronal stimulation with optogenetics, while the in vivo SCN receives light input through the retina and feedback inputs from other brain regions. Also, the behavioral output is downstream of the SCN clock gene rhythms. The rate of entrainment varies depending on the strength of zeitgeber (e.g., light input) as exemplified in the phase response varying with different light intensities (Johnson, Elliott and Foster. Chronobiol. Int. 2003). Thus, the rate of entrainment for the ex vivo SCN rhythms could be different from that for behavioral rhythms. Indeed, our results in the figure 4A and Figure 4–supplement 3 suggest that the ex vivo SCN rhythms entrain more rapidly to the 16:8 skeleton photoperiod than do behavioral rhythms.

2) It is strongly suggested that the authors use the same phase markers in all experiments and at least one additional phase marker, preferably 'center of gravity' which effectively incorporates changes in waveform shape. Suggestions also include measuring period with an additional method (particularly as only a few days are measured for the analyses), as well as considering other interpretations for some of their observations (how spatial expression of ChR may explain the differences in dSCN and vSCN PER2 patterns following optostimulation). Importantly, to rule out the latter they should quantify the fraction of SCN cells that express their optogenetic driver.

We understand the reviewers’ concerns. In all experiments we used the peaks for SCN rhythms in free-running conditions “prior to” and “following” optogenetic light cycles, while, we used the half-maxes of the rising phases for SCN rhythms “during” entrainment by light cycles because waveform changes following stimulation sometimes make it difficult to determine the peaks, especially when the waveform changes occur around the peak time. We kept this rule consistent across all conditions and made comparisons between the same phase markers across experiments. To better reflect any waveform changes during entrainment, we have now included acrophase, the peak of a sine wave fitted to data, as another phase marker which models center of mass (Figure 3–supplement 3, Figure 4–supplement 1), which also illustrates stable phase entrainment. To address a concern over the period measurement, we have now analyzed our data using another method based on different assumptions, the Lomb-Scargle periodogram (Figure 2F right, new Figure 3–supplement 2, new Figure 4–supplement 2), and find essentially identical results. To test whether regional heterogeneity of phase shifts in the SCN is attributed to spatial expression pattern of the optogenetic construct ChrimsonR, we measured from existing imaging data the subregional ChrimsonR expression in the SCN used in the regional analysis in Figure 6. We compared the ChrimsonR expression between the clusters of different phase shift magnitude and found that clusters with a larger phase shift magnitude in fact had had a lower ChrimsonR expression on average (new Figure 6–supplement 1). This suggests that regional increases in the phase response are not artifacts from increased optogenetic construct expression. We have added the text accordingly as shown below:

“To test whether a higher regional expression of the ChrimsonR optogenetic construct in the lateral or ventrolateral SCN might be associated with the larger phase shifts of this region, we analyzed the ChrimsonR-tdT fluorescence intensity in different subregions corresponding to phase-shift clusters and compared with the magnitude of phase shifts. For both CT14 and CT21 stimulations, subregions exhibiting large phase shifts had lower levels of ChrimsonR-tdT expression (Figure 6—figure supplement 1), suggesting that regionally different phase responses are not directly derived from spatial ChrimsonR expression patterns. Given this distribution, and that we applied synchronous stimulation across the entire SCN, our results suggest that this subregional heterogeneity in resetting responses is intrinsic to the neurons or circuits in the ventrolateral and dorsomedial SCN.” (Results, pp. 9-10, lines 367-377)

3) Modify several aspects of the text (avoid excessive clock-terms/jargon), clarify some figures (i.e. Figure 4), and explain the rationale behind the choice of T-cycles and photocycles that were tested, to help a general audience to better understand key aspects of the work.

We have removed jargon, including “transforms the waveform … into a highly asymmetrical trajectories”, and described them in more general terms. We have made changes in the figures, including applying different colors to dawn and dusk pulses in Figures 4 and 5, and adding the photoperiod details in Figure 5.

We applied one-pulse T-cycle entrainment because it is a classic and simplest form of entrainment experiment so that we can easily interpret results of whether ex vivo SCN can really entrain to periodic optogenetic stimulation. We have explained the rationale behind using one-pulse T cycles as shown below.

“Individual phase shifts by external time cues are the building blocks of circadian entrainment (Pittendrigh and Daan, 1976b). Daily, repeated phase shifts to light adjust the endogenous circadian period to match the 24h day and align circadian rhythms in a particular temporal relationship with the daily light-dark cycle (i.e., phase angle of entrainment). Remarkably, animals can entrain to single light pulses given repeatedly as a cycle (T-cycle), even to cycle periods that deviate modestly from 24 hours, such as a 22h light cycle (Pittendrigh and Daan, 1976b). This so-called one-pulse entrainment is the simplest form of entrainment well-documented in the literature and can be used to easily interpret whether biological clocks can entrain to certain external cues.” (Results, p. 5, lines 179-187)

To detect the real-time dynamics of the clock gene rhythm waveforms encoding seasonal variation of daylight lengths, we applied skeleton photoperiod entrainment, consisting of dawn and dusk pulses every day as it mimics most aspects of a full photoperiod entrainment in vivo (Pittendrigh and Daan. J. Comp. Physiol. 1976). The rationale behind using skeleton photoperiods can be found in the first paragraph of a section describing the skeleton photoperiod experiments.

“Circadian oscillators in the SCN encode the length and timing of the daily light period (i.e., photoperiod) (Goldman, 2001; Rusak and Morin, 1976), thereby promoting seasonal changes in physiology and behavior. Strikingly, light cycles consisting of only brief light pulses defining dawn and dusk (i.e., skeleton photoperiods) have been shown to simulate most aspects of full photoperiods at the behavioral level, with the interval between the brief dawn and dusk pulses determining the photoperiodic state of circadian behavior (Pittendrigh and Daan, 1976b). This remarkable ability of the circadian system to lock onto the timing of light transitions can, in principle, result from properties of circadian photoreception in the retina (input), downstream behavioral modulation (output), or properties of the SCN clock itself. Here we have tested directly whether the SCN clock itself can be entrained to different photoperiods by brief daily transitions that simulate dawn and dusk.” (Results, p. 7, lines 225-235)

We also described in the discussion as a technical limitation that our current system is not readily applicable to a full light cycle entrainment.

4) The authors should seek better evidence that optogenetic stimulation did not kill cells, impair redox state, or PER2 cycling.

We clearly showed in Figure 1 that changing the light exposure from blue to red light significantly preserves the rhythmic amplitude of PER2 clock gene following 12 hours of light exposure. The amplitude of PER2 cycling following 12h of red light was not statistically different than that of the sham control, whereas PER2 cycling was undetectable following 12h of blue light. To further minimize any potential phototoxicity by red light in our experiments, we limited red light exposure to between 15 mins to 1.5 hours duration – only 2-12% of the 12h test exposure in Figure 1 – and were able to record robust PER2 cycling during and following 9 days of successive stimulation (Figure 4). We consider this reasonable evidence that red light stimulation, as we have employed it in our experiments, does not impair the PER2 rhythms we measure. Finding the exact nature of the induced phototoxicity by blue light, which we did not use beyond the initial test in Figure 1, is beyond our scope.

5) The authors should address the concern that phase shifts were calculated immediately after perturbation and compare their findings to any steady-state phase shifts after correcting for any induced period changes (see Rev #2, point 4).

We understand the reviewer’s concern. With regard to the calculation of phase shifts – we calculated phase shifts (Figures 2 and 6) using a standard method in the field: using linear regression of multiple cycles pre and post stimulation to derive the difference in hours between the actual post-stimulation peaks and the peaks predicted from pre-stimulation cycles (Mazuski, et al., PMC6085153; Patton, et al., PMC7341843). By using regression of period prior to and following the stimulation, this method takes into account period changes following stimulation in calculating the phase shift. Example regression lines and calculated phase shifts are shown in Figure 2A. Also, we did not observe abrupt cycle-to-cycle period changes, an indication of transients, except on the single directly stimulated cycle, nor did we observe ongoing waveform changes in the free-runs following acute stimulation that might affect calculation of phase shifts. In addition, Reviewer 2 asks in comment 4 if there are methodological differences between the phase shifting data in Figures 2 and 6 that could explain the differences in observed amplitude. Indeed, these experiments were performed in different apparatus using different light sources, which could explain these differences, as we further explain below. Phase shifts and period changes were calculated using the same methods in Figures 2 and 6.

The phase data presented in Figures 3 and 4 are steady-state phase angles to ongoing repeated daily entraining pulses, rather than phase shifts per se as in Figures 2 and 6. We have addressed the concerns regarding waveform changes during entrainment by displaying acrophases, which also show steady state phase angles of entrainment (Figure 3–supplement 3, Figure 4–supplement 1).

Reviewer 2 suggests in comment 4 that calculating the phase shifts a different way would allow us to compare with in vivo behavioral phase shifts. However, the method we have used to calculate our ex vivo phase shifts is indeed the most standard method for calculating behavioral phase shifts as well, already allowing this comparison. In addition, there are other factors (e.g. zeitgeber stimulus intensity) that also contribute to the magnitude of phase shifts (Johnson, et al., doi.org/10.1081/CBI-120024211) and may be difficult to compare directly between in vivo and ex vivo experiments. Clearly, our results demonstrate that excitatory optogenetic stimulation of the SCN ex vivo produces phase shifts of similar magnitudes and directions as in vivo light stimulation, but more precise comparisons are tenuous at this point in our opinion.

Reviewer #1:The work by Kim and McMahon describes an exciting combination of methods that allow obtaining accurate circadian data from suprachiasmatic nucleus (SCN) slices over many days, while exploring how key clock parameters are affected by perturbations mimicking regular light cues, ex vivo. In order to do so, the authors adopt an optogenetic strategy based on ChrimsonR: a light-gated cation channel that upon red-light activate neurons. The choice of red-light over blue was not casual, as the latter can have negative effects on the primary culture if prolonged stimulation is used. Previous studies that have focused on the effect of different photoperiods or light-perturbations on core-clock dynamics in the SCN were either indirect measurements (running-wheel activity), had limited resolutions (i.e. SCN samples collected from different animals every 4 h), or implied monitoring real-time reporters in SCN slices, after the particular entrainment or perturbation had occurred in the animals. With the new approach presented by Kim and McMahon it is possible to monitor any reporter of choice (in this case PER2LUC), and see what happens in response to particular light perturbations (by mimicking neural firing through the optogenetic system). Thus, the authors are now able to obtain accurate information on PER2LUC dynamics upon defined perturbations, that they dissect focusing on changes in oscillations waveforms: they analyze what happens in terms of phase and period when discrete perturbations (designed to mimic a 15 min light pulse seen by the animal) are given at different times throughout 24 h (reproducing cycles of different T). The analyses also included exploring the effect of skeleton cycles of different photoperiods. Notably, they can reproduce with this ex-vivo perturbations several circadian aspects described in foundational papers (i.e. Pittendrigh and Daan, 1976), while also observing phenomena that are different from what one may expect, reflecting more complex organismal responses. Thus, among some of the interesting findings is that the ability (or bias) to resolve ambiguous skeleton light cycles favoring short-day entrainment, appears to reside in the SCN clockworks, as deduced from their ex-vivo work. Also, their observations indicate that repeated stimulation does not produce large period after-effects, whereas single light stimulation does so. Finally, their approach also allowed them we observe different responses in lateral or ventrolateral vs medial or dorsomedial SCN changes in phase or period responses revealing an intrinsic nature of the clock-resetting heterogeneity in the SCN.While the work is interesting, it is key to enrich the analysis and discussion of the data under the light of other existing publications that have utilized similar approaches (i.e optogenetic perturbation affecting SCN firing and clock dynamics), or that have pursued addressing similar questions regarding light entrainment. Thus, the latter includes distinct publications analyzing SCN slices coming from animals kept under diverse photoperiods, or in vivo monitoring (in freely moving animals) of the SCN (via clock reporters or electrophysiological recorders) of animals subjected to different light stimulation protocols..

Thanks for your support of our work in the manuscript.

In addition, it becomes important to better explain for a general audience the choice of how T-cycles and photocycles were established i.e. with single light pulses or skeleton photoperiod as opposed to classic LD cycles. Indeed, one of the advantages of implementing this red-light optogenetic system is that prolonged light stimulation can be applied.Therefore it is puzzling to see that the authors only used it for discrete 15 min light pulses as opposed to light (red-light) mimicking full photoperiods and not just skeleton ones..

We applied one-pulse T-cycle entrainment because it is a classic and simplest form of entrainment so that we can easily interpret results of whether ex vivo SCN can really entrain to periodic optogenetic stimulation. We have now added more explanations for the rationale behind doing this paradigm.

“Individual phase shifts by external time cues are the building blocks of circadian entrainment (Pittendrigh and Daan, 1976b). Daily, repeated phase shifts to light adjust the endogenous circadian period to match the 24h day and align circadian rhythms in a particular temporal relationship with the daily light-dark cycle (i.e., phase angle of entrainment). Remarkably, animals can entrain to single light pulse given repeatedly as a cycle (T-cycle) such as a 22h light cycle (Pittendrigh and Daan, 1976b). This so-called one-pulse entrainment is the simplest form of entrainment well-documented in the literature and can be used to easily interpret whether biological clocks can entrain to certain external cues.” (Results, p. 5, lines 179-187)

The rationale behind using skeleton photoperiods can be found in the first paragraph of a section describing the skeleton photoperiod experiments.

“Circadian oscillators in the SCN encode the length and timing of the daily light period (i.e., photoperiod) (Goldman, 2001; Rusak and Morin, 1976), thereby promoting seasonal changes in physiology and behavior. Strikingly, light cycles consisting of only brief light pulses defining dawn and dusk (i.e., skeleton photoperiods) have been shown to simulate most aspects of full photoperiods at the behavioral level, with the interval between the brief dawn and dusk pulses determining the photoperiodic state of circadian behavior (Pittendrigh and Daan, 1976b). This remarkable ability of the circadian system to lock onto the timing of light transitions can, in principle, result from properties of circadian photoreception in the retina (input), downstream behavioral modulation (output), or properties of the SCN clock itself. Here we have tested directly whether the SCN clock itself can be entrained to different photoperiods by brief daily transitions that simulate dawn and dusk.” (Results, p. 7, lines 225-235)

In summary, to detect the real-time dynamics of the clock gene rhythm waveforms encoding seasonal variation of daylight lengths, we applied skeleton photoperiod entrainment, consisting of dawn and dusk pulses every day as it mimics most aspects of a full photoperiod entrainment in vivo (Pittendrigh and Daan. J. Comp. Physiol. 1976).

It is certainly exciting if we could get SCN rhythms ex vivo throughout a full light cycle. However, we did not follow up on this on our manuscript because our current system cannot record SCN rhythms while SCN slices are stimulated. Bioluminescence recording is interfered with by the excitation light for optogenetics. Also, the photomultiplier tubes (PMTs) in the Lumicycle are very light-sensitive and can break down if exposed to strong light that we use for optogenetics, so we had to turn off the PMTs when we stimulated SCN slices. In the case of a full light cycle, data gaps would arise for the entire light phase, which makes it very difficult to track SCN rhythms. Also, there is a small effect on the amplitude of rhythmicity even with a red light if it is a long exposure (Figure 1B), so we wanted to take caution.

1 – "We found that PER2::LUC bioluminescence became arrhythmic following the prolonged blue light exposure (Figure 1A, B). This effect was not reversible with a medium change (Figure 1A), suggesting that long-term blue light exposure per se can elicit photodynamic damage of SCN slice cultures".The authors utilized (470nm, 10Hz frequency, 10ms width, 1.2mW/mm2) for 12h. Could they comment whether (i) lower intensity (or change in the duty cycle) could be used (to obtain similar response if they were containing an optogenetic construct)

We did not test whether changing the light intensity or the duty cycle could mitigate the blue light-induced phototoxicity ex vivo. The light intensity we used is only slightly above the minimum intensity for opsin activation (~1mW/mm^2^), so it could not be reduced. The duty cycle is 10% (=10ms pulse every 100ms). This again is near the minimum duration (10ms) to reliably evoke a spike in an SCN neuron. So while decreasing these two factors might improve slice health following blue light exposure, stimulation would likely not successfully drive optogenetic activation of the neurons.

2) Better define and elaborate what the "photodynamic damage" may be. i.e is it possible to use any marker that would be informative of apoptosis, mitochondrial damage, DNA fragmentation etc?

Given that blue light-illuminated SCN slices did not recover from arrhythmicity and the rhythm baseline stayed the same after the culture medium was changed, this indicates that the photodynamic damage or the phototoxicity is irreversible. It may include cell death and mitochondrial damage. The exact nature of the phototoxicity to blue light is beyond the scope of our study as we have concentrated on using red light which does not evoke these problems in our use.

2 – The authors ought to discuss their results in the context of other relevant work that has used similar approaches or pursued similar questions: i.e. Patton et al., PMID: 32636383, utilizing ChR to control with blue-light firing of VIP and VPAC2 neurons; or Mei et al., PMID: 29610316 following Per2 and Cry1 in vivo dynamics (by monitoring reporters in freely moving animals) under some particular light conditions; or the work of VanderLeest et al., PMID: 17320387 conducting in vivo electrophysiological recordings in animals under short or long days., among other papers that may be addressing similar questions.

Thanks for your suggestions. We have added in the discussion the text as below.

“Light exposures at earlier dawns and later dusks for long photoperiods drive more PER2 induction than do light exposures in equinox photoperiods, thus expanding PER2 waveform. This is consistent with waveform broadening of neural activity rhythms in animals under a long photoperiod (VanderLeest et al., 2007).” (Discussion, p. 11, lines 437-440)

“Decreased phase synchrony among SCN neurons following light exposure has been observed in the case of VIP-induced phase shifts (Hamnett et al., 2019) and entrainment to long photoperiods or non-24h light cycles (Azzi et al., 2017; Evans et al., 2013). Interestingly, stimulating VIP neurons reset the ensemble SCN phase, while stimulating VPAC2 neurons located in the dorsal SCN do not (Patton et al., 2020), suggesting regionally different phase-resetting capacity.” (Discussion, p. 12, lines 502-507)

“Interestingly, real-time clock gene rhythms in the in vivo SCN re-entrain to a new light cycle during experimental jet lag (Mei et al., 2018). Future waveform analyses of clock gene rhythms in such an in vivo setup would allow comparison with our observations in the isolated SCN.” (Discussion, p. 13, lines 519-522)

3 – Figure 1: ChrimsonR-tdT is mentioned in the figure legend, yet, it is not properly explained in the manuscript, although the ChrimsonR is at the heart of the optogenetic system utilized in this work.

Thank you for pointing this out. We have now revised the text to:

“We thus expressed a red light-activated opsin, ChrimsonR, fused with a red fluorescent protein tdTomato (tdT) throughout SCN slices using synapsin promoter-controlled AAVs targeting all SCN neurons (AAV-Syn-ChrimsonR-tdT, Klapoetke et al., 2014) to mimic widespread retinal photic inputs to the SCN (Chen et al., 2011; Fernandez et al., 2016) (Figure 1C).” (Results, p. 3, lines 112-116)

4 – Supplement 1. It is nice to evidence the major effect that stabilized glutamine, and utilizing slices from younger mice have on rhythms. There are a couple of things that the authors could further comment on this:a) There seems to be a slight phase delay in the P12 -Ala-Gln traces. Could the authors comment on this?

We did not see a consistent phase delay in the P12 Ala-Gln. We think the phase difference in this example is due to the natural variation of PER2 rhythm phase or period across animals.

b) The great improvement in signal robustness and amplitude results from changing two variables at once (age of the animals source of SCNs and media composition). Therefore, it is hard to weight on the relative contribution of each variable to rhythm quality. Therefore, if the authors have data allowing to understand which variable is more critical it would be great to include that in the manuscript. Importantly, other groups doing SCN explants analyses also use young animals (i.e P8-P10 PMID: 32636383; P3 PMID: 25186748), or already use glutamax (see below). Therefore, what would be the added value/novelty of the presented data (Figure 1, suppl. 1)?

We agree but do not have enough data to tell which variable is more critical. We think each variable is potentially important. Younger brain slices have a better survival in culture generally, as has been widely demonstrated, and ammonia byproduct buildup in such a static culture would be more detrimental over time. Although we are not first to use either slices from young animals or glutamax for SCN culture, we decided to add the data in the supplement because improving long-term slice viability was critical to assaying SCN rhythms throughout our extended ex vivo entrainments.

c) The authors use glutamax 0.2 mM, whereas other experiments (i.e. Patton et al., PMID: 32636383) use 2 mM glutamax, also achieving good readings for well over 15 days.

Thank you for finding this error. We used 2mM glutamax as per manufacturer’s recommendation and have corrected this in the manuscript.

5 – Figure 1—figure supplement 2. The system that the authors have elaborated to optogenetically control their samples while also monitoring PER2-LUC expression is extremely useful and for sure will be a valuable resource for other colleagues in the field. Therefore, it is recommended that they provide more technical details of the setup (actual photos of the setup, radius of the illuminated region etc), and make available the Matlab code used to run the hardware/retrieve the luc data.

Thank you for the positive assessment of our system. The actual setup is well depicted in Figure 1D, Figure 1-supplement 2, and corresponding texts in the methods section. We have added the illumination radius (6mm) in the text. We have deposited the Matlab code to GitHub (link below). We are discussing developing this system with the Lumicycle manufacturer Actimetrics to make it available to other colleagues in robust form. https://github.com/SuilKim/OptoLumicycle

6 – Figure 2A. It appears that the traces corresponding to the CT21 intervention are showing an altered period from even before the stimulation. Could the authors check and comment on that?

Thank you for your detailed observation. Period lengths across the example traces before stimulation are in the range of period variability found across animals in our colony, and they are not significantly different.

7 – Page 8: "Further, the acute phase shifts were accompanied by a subsequent sustained change in the free-running circadian period". From the data it is not fully clear:1) When (after starting the SCN cultures) was red-light stimulation applied (from Figure 2 it seems that on the 3rd day of recording). Please explicitly mention it in the main text.

We applied the stimulation on the 3^rd^ or 4^th^ day of recording. We have added text:

“we applied 10Hz optogenetic light stimulations to ChrimsonR-expressing SCN slices on the 3rd or 4th day of recording at three different circadian times (CT) as defined by the timing of the intrinsic PER2::LUC rhythm of each SCN slice.” (Results, p. 4, lines 134-137)

2) How was the period calculated (based on how many days of recording after stimulations; the authors only mention "several days"). This is important, to make sure that the calculated changes in period do not correspond to, nor are affected by, transients. This becomes quite relevant as subsequent experiments involving repeated perturbations do not show period effects.

We calculated the pre- and post-stimulation periods using three or four days of recording before and after stimulation, respectively. We did not see abrupt changes in the cycle-to-cycle period after stimulation, an indication of transients, in our experiments except for the one stimulated cycle.

8 – Page 8: "Stimulation at CT14 elongated the falling phase to induce a phase delay"Another way to describe what is seen is that stimulation causes a new rise, delaying the falling phase and therefore causing a phase delay.

We have changed the text …to:

"Stimulation at CT14 induced an acute rise in PER2, elongating the falling phase to induce a phase delay" (Results, p. 4, lines 156-157)

9 – Page 8: "whereas stimulation at CT21 prematurely ended the falling phase and accelerated the rising phase to induce a phase advance".Looking at the data it makes it harder to see such a "premature end" of the falling phase, as the stimulus is given around the time of the trough. I agree that it accelerates de rising though, leading to a phase advance.

We have changed the text …to:

"whereas stimulation at CT21 accelerated the rising phase to induce a phase advance" (Results, p. 4, lines 157-159)

10 – Page 8: Together, differential waveform changes in PER2 rhythms induce time-dependent phase responses to light stimulation in the SCN".The way this phrase is constructed seems to implicitly denote causation on the waveform properties of Per2, where actually a more accurate description would be related to the changes on the state variable itself.

Thanks for pointing this out. We agree with the reviewer. We have now changed the text to:

“Together, differential waveform changes in PER2 rhythms underpin time-dependent phase responses to light stimulation in the SCN." (Results, p. 4, lines 161-163)

11 – Figure 3E. The traces corresponding to the "before" T22 or T25 cycles (black lines) appear to be different in both experimental sets. Thus, while in the left graph the pre-T curve has a peak centered around 18, in the right graph the peak is centered around 15 or so.

Thanks for asking this question. Representative traces have different peak times due to different endogenous periods. This is a natural variance in the period between SCN slices.

12 – It is not clear why the authors chose for the entrainment protocol of different T only 1 light pulse, instead of opting for a skeleton photoperiod (although they did use skeleton photoperiod in following experiments). Moreover, based on the improved properties of this red-light optogenetic systems the authors could have even tried subjecting the SCN slices to full light:dark cycles of different T (i.e 12 h light, 12 dark), as one of the points of utilizing a red-light optogenetic system, was to actually be able to activate the system for many hours (with no cytotoxicity). Could the authors comment on the logic behind their choice and the limitations of only exploring T cycles pushed with one light pulse (as opposed to skeleton or full photoperiods) ?. Likewise, their system, as opposed to the ones relaying on blue-light, allows creating more realistic LD regimes (i.e full 12 h of light, as opposed to a skeleton one).

Thanks for asking this question! We applied one-pulse entrainment because it is a classic and simplest form of entrainment so that we can easily interpret results of whether ex vivo SCN can really entrain to periodic optogenetic stimulation. One-pulse entrainment is sufficient to test whether ex vivo SCN can entrain to a different T cycle. It is certainly exciting if we could get SCN rhythms ex vivo throughout a full light cycle. However, we did not follow up on this on our manuscript because our current system cannot record SCN rhythms while SCN slices are stimulated. Bioluminescence recording is interfered with by the excitation light for optogenetics. Also, the photomultiplier tubes (PMTs) in the Lumicycle are very light-sensitive and can break down if exposed to strong light that we use for optogenetics, so we had to turn off the PMTs when we stimulated SCN slices. In the case of a full light cycle, data gaps would arise for the entire light phase, which makes it very difficult to track SCN rhythms. Also, there is a small effect on the rhythmicity even with a red light if it is a long exposure (Figure 1B), so we wanted to take caution.

13 – Page 18: "Group averaged clustering analysis identified a lateral-medial axis for phase shifts and a ventrolateral-dorsomedial axis for period changes. In terms of changes in phase coupling, the phase variance within the SCN was increased following stimulation (Figure 6B), indicating a larger phase desynchrony in the SCN."This is an interesting inference from the data. Could the authors comment more on how this compares with prior SCN-luc section studies derived from animals kept under different light regimes?

Our data showed that acute light stimulation decreases a phase synchrony in the SCN. Similarly, applying vasoactive intestinal peptide (VIP), a neuropeptide involved in light-induced clock resetting, in the SCN causes a reduced phase coherence between cells (Hamnett, et al., PMC6358603). In the case of light entrainment, entraining mice to long photoperiods (Evans, et al., PMC3841113) or non-24h light cycles (Azzi, et al., PMC5247339) causes a larger phase variance in SCN rhythms ex vivo. We now have added the text in the discussion.

“Decreased phase synchrony among SCN neurons following light exposure has been observed in the case of VIP-induced phase shifts (Hamnett et al., 2019) and entrainment to long photoperiods or non-24h light cycles (Azzi et al., 2017; Evans et al., 2013). Interestingly, stimulating VIP neurons reset the ensemble SCN phase, while stimulating VPAC2 neurons located in the dorsal SCN do not (Patton et al., 2020), suggesting regionally different phase-resetting capacity.” (Discussion, p. 12, lines 502-507)

14 – Page 20: " we directly showed that discrete light input induces acute induction of PER2 expression and differential waveform changes in PER2 rhythms in a phase-dependent manner, leading to a phase-dependent resetting response.."Instead of talking about resetting, wouldn't it be more appropriate to talk about phase shifts (at least for the intensity of the utilized light-pulses)?

We agree with the reviewer. We have now changed the text to:

“We directly showed that discrete light input induces acute induction of PER2 expression and differential waveform changes in PER2 rhythms in a phase-dependent manner, leading to time-dependent phase shifts.” (Results, p. 10, lines 405-408)

15 – Materials and methods:– Could the authors comment on why they used heterozygous (as opposed to homozygous) PER2::LUC mice? Were both male and female individuals used for SCN preparations?

We used heterozygous PER2::LUC mice as the PER2::LUC knock-in allele can alter circadian functions such as free-running period and entrainment (Ralph, et al., PMC8191895). We used both males and females for our experiments. We have added the text in the methods.

“P11-14 heterozygous PER2::LUC knock-in mice (Yoo et al., 2004) were used for organotypic slice culture and subsequent procedures as the PER2::LUC knock-in allele can alter circadian functions such as free-running period and entrainment (Ralph et al., 2021). All animals were housed in a 12:12 LD cycle, and had food and water provided ad libitum. Both male and female mice were used in experiments.” (Methods, p. 14, lines 542-546)

– "Custom-written code in Matlab (Mathworks) was used to access luminescence data collection software (Actimetrics), the multifunctional I/O device, and a signal generator software (Mhinstek)." Available upon request?

We have deposited the Matlab code to GitHub (link below). We are discussing developing this system with the Lumicycle manufacturer Actimetrics to make it available to other colleagues in robust form.

https://github.com/SuilKim/OptoLumicycle

Reviewer #2:The authors introduce a new system to record clock gene expression (as luminescence) for weeks while using optogenetic stimulation to test whether the isolated suprachiasmatic nucleus (SCN) exhibits circadian properties of period aftereffects, entrainment to non-24 h cycles and to skeleton light cycles previously described in vivo. They present the results from well-designed experiments which validate the technique and reveal that these properties are indeed intrinsic to the SCN slice. It is nice to see classical chronobiology concepts revisited and attributed to changes in gene expression of a small neural network. The major conclusions include: rapid optogenetic activation of SCN neurons results in changes in SCN PER2 waveform that correlate with changes in behavior induced by seasonal or non-24h environmental cues. This manuscript provides exciting results with excellent experimental design. The relevance of rapid changes in firing to rapid changes in PER2 and, ultimately, long-term changes in phase and period could be made more clear for a general readership. The manuscript will be improved when the authors address some major concerns with the analysis.1) The authors conclude that each feature of circadian behavior that they sought to attribute to the SCN was, indeed, found in the isolated SCN. The authors should consider whether there are features that differ between the behavior and the isolated SCN. For example, did the rate of entrainment in vivo differ from in vitro? Importantly, to reach their conclusions, they use different phase markers – peak and mid-point of rising phases- in different places in the manuscript. Was this necessary to reach their conclusions?

Thanks for asking the questions. The rate of entrainment depends on the strength of zeitgeber as exemplified in the phase response varying with different light intensities (Johnson, et al., doi.org/10.1081/CBI-120024211). Light signals in vivo are transmitted to the SCN through the retina, while we used optogenetics to directly stimulate SCN neurons. Also, the behavioral output is downstream of SCN clock gene rhythms. Thus, we think that the rate of entrainment ex vivo is not necessarily the same as that in vivo. If you compare the 16:8 photoperiod between in Figure 4A and Figure 4–supplement 3, you can see that ex vivo SCN rhythms entrain more rapidly than do locomotor behavior rhythms.

We used as a phase marker the peaks of SCN rhythms in free-running conditions before and after light cycles when the PER2::LUC waveform is essentially sinusoidal. However, we used the half-maxes of the rising phases for SCN rhythms during entrainment by optogenetic light cycles because the induced waveform changes from stimulation sometimes make it difficult to determine the peaks, especially when the waveform changes occur around the peak time. We kept this rule consistent across all experiments.

2) To measure changes in period and phase, the authors should justify their choice for phase marker and why they change this for some experiments. In the context of this manuscript, the change in waveform shape can lead to over-estimation of phase and period differences [cf., Daan S, Oklejewicz M. The precision of circadian clocks: assessment and analysis in Syrian hamsters. Chronobiol Int. 2003].

In the previous answer, we covered the reasons why we have applied the phase markers across all experiments.

We suggest that the authors use the same phase markers in all experiments and at least one additional phase marker, preferably 'center of gravity' which effectively incorporates changes in waveform shape. Observing similar results with 'center of gravity' gives a better picture of the magnitude of effects in response to optogenetic stimulation that the authors report here. [Kenagy, G. J. 1980. Center-of-gravity of circadian activity and its relation to free-running period in two rodent species. J. Interdiscip. Cycle Res.].

We agree and explain above that we have used the same phase markers across all experiments. We also agree that adding a phase marker for the temporal center of PER2::LUC bioluminescence would be informative. We now have added the acrophases in the Figure 3–supplement 3 and Figure 4–supplement 1, which approximate the center of gravity (Díez-Noguera, PMC3723718). We added in the methods section “Acrophases were calculated using ClockLab software.” (Methods, p. 16, line 592)

Because the authors use only 3 days of data to estimate the circadian period, they should use at least one additional method to measure the period, preferably a method like wavelet, periodogram or FFT which has different assumptions. [cf., Maria J. Costa, Bärbel Finkenstädt, Véronique Roche, Francis Lévi, Peter D. Gould, Julia Foreman, Karen Halliday, Anthony Hall, David A. Rand, Inference on periodicity of circadian time series, Biostatistics, Volume 14, Issue 4, September 2013, Pages 792-806. Also: Zielinski T, Moore AM, Troup E, Halliday KJ, Millar AJ (2014) Strengths and Limitations of Period Estimation Methods for Circadian Data. PLOS ONE 9(5): e96462.]

Thanks for pointing this out. We have now added the period quantification using the Lomb-Scargle periodogram in the Figure 2F right, Figure 3–supplement 2, and Figure 4–supplement 2. We added in the methods section:

“Period length was calculated using a linear regression of at least three peaks or using Lomb-Scargle periodogram in ClockLab software.” (Methods, pp. 15-16, lines 589-591)

3) Figure 1 can be moved to supplementary results and, in fact, raises some concerns. The major point of this methodological figure is that the treatment (red optogenetic stimulation) is non-toxic, but the results are not convincing. With only three replicates, it appears the amplitude was more variable and trending to reduced following red light stimulation. The authors should seek better evidence that optogenetic stimulation did not kill cells, impair redox state, or PER2 cycling.

With respect to the Reviewer, we would like to keep Figure 1 as a main figure. We think that it is informative to general readers as we address a critical issue, phototoxicity, in using optogenetics for ex vivo entrainment and introduce our device setup used to generate data shown in later figures.

Indeed, there are small effects on rhythms amplitude even with red light in this extreme example using 12-hour stimulations. However, our data strongly supports that switching blue light to red significantly improves SCN rhythmicity to a degree close to the sham control, and in our actual experiments we limited stimulations to 15 mins to 1.5 hrs duration – 2-12% of the 12 hour exposure that with red light produced a barely detectable decrease in amplitude. Finding the exact nature of the induced phototoxicity of blue light, which we did not use beyond the initial test, is beyond our scope.

4) Figure 2 validates what has been shown by this lab and others: Optogenetic stimulation of the SCN in vitro or in vivo can shift circadian rhythms. The authors should address the concern that phase shifts were calculated immediately after perturbation and compare their findings to any steady-state phase shifts after correcting for any induced period changes. This includes changes to the measurements in Figures 3, 4 and 6. This will allow the authors to: 1) measure similarities between in vitro and in vivo phase shifting and 2) quantify how much of the shift is due to a period change vs. an instantaneous phase shift. For example, the authors report large magnitude (4-5h) phase-shifts following stimulation at CT14 and CT21 in Figure 1. However, in Figure 6A most ROIs indicate a phase shift of only about 2h and very few ~3h. Similarly, in 6B, only a small proportion of ROIs indicate 4+h phase shift. Is this the result of differences in how the data were collected or analyzed?

Thanks for asking these questions. We calculated phase shifts (Figure 2) using a standard method in the field: using linear regression of multiple cycles pre and post stimulation to derive the difference in hours between the actual post-stimulation peaks and the peaks predicted from pre-stimulation cycles (Mazuski, et al., PMC6085153; Patton, et al., PMC7341843). This method takes into account period changes following stimulation in calculating the phase shift. Example regression lines and calculated phase shifts are shown in Figure 2A. Also, we did not observe abrupt cycle-to-cycle period changes, an indication of transients, except on the directly stimulated cycle, nor did we observe ongoing waveform changes during the free-runs following acute stimulation that might affect calculation of phase shifts.

The method we have used to calculate our ex vivo phase shifts is indeed a standard method for calculating behavioral phase shifts as well, already allowing this comparison. In addition, there are other factors (e.g. zeitgeber intensity) that also contribute to the magnitude of phase shifts (Johnson, et al., doi.org/10.1081/CBI-120024211) and may be difficult to compare directly between in vivo and ex vivo experiments. Clearly our results demonstrate that excitatory optogenetic stimulation of the SCN ex vivo produces phase shifts of similar magnitudes and directions as in vivo light stimulation, but more precise comparisons are tenuous at this point.

For differences in the phase shift magnitude between Figures2 and 6 – Figure 2 data was collected in the Lumicycle luminometer, while Figure 6 data was collected in the LV200 microscope. We used the same light pulse frequency and duty cycle, but different optics (e.g., lenses), a slightly lower light intensity (~1.1mW/mm^2^ vs. ~1.5mW/mm^2^), and a slightly longer light wavelength (635nm vs. 625nm) were used for optogenetic stimulation in the LV200. The differences in stimulation in the two apparatus may have resulted in different magnitude phase shifts. Phase shifts and period changes were calculated in the same way.

With regard to whether phase data in Figures 3 and 4 should also be re-examined for steady-state vs. transient phase shifts – the phase data presented in Figures 3 and 4 are in fact steady-state phase angles to ongoing repeated daily entraining pulses, rather than phase shifts per se. We have addressed the concerns regarding waveform changes and determination of phase by using acrophases (Figure 3–supplement 3, Figure 4–supplement 1) as well as the previously used markers.

5) Can the authors relate the spatial (throughout the SCN) and temporal (10Hz, 10ms pulses) optogenetic stimulation to photic activation of the SCN? To begin to do this, the authors should quantify the fraction of SCN cells that express their optogenetic driver. They should make it clear in the Results that they depend on viral transduction and the synapsin promoter (appears on page 22 of the manuscript while the tool is used starting on page 3). Presumably this results in neuron-specific expression of their ChrimsonR. For example, does the spatial expression of ChR explain the differences in dSCN and vSCN PER2 patterns following optostimulation (Figure 6)?

Thanks for your suggestion. We have added more descriptions about how our optogenetic stimulation are related to light activation of the SCN in vivo and the viral construct we used. Regarding ChrimsonR expression pattern in the SCN, ChrimsonR (ChR) expression driven by the synapsin promoter is widespread across the SCN slice as seen in the figure 2C. We had pre-stimulation ChR-tdTomato fluorescence imaging data for the same slices used for regional analysis in Figure 6. Thus, we were able to quantify the subregional expression of ChR, and then the correlation between ChR expression and phase shift magnitude across different phase-shift clusters. We found that SCN regions demonstrating higher magnitude phase shifts (e.g. ventrolateral regions) actually had lower levels of ChR expression relative to regions exhibiting lower magnitude phase shifts (e.g. dorsomedial regions) as shown in the figures below. Thus, this suggests that regional increases in the phase response are not artifacts from increased optogenetic construct expression. We have added the figures in the Figure 6–supplement 1. We edited the text to:

“To test whether a higher regional expression of the ChrimsonR optogenetic construct in the lateral or ventrolateral SCN might be associated with the larger phase shifts of this region, we analyzed the ChrimsonR-tdT fluorescence intensity in different subregions corresponding to phase-shift clusters and compared with the magnitude of phase shifts. For both CT14 and CT21 stimulations, subregions exhibiting large phase shifts had lower levels of ChrimsonR-tdT expression (Figure 6—figure supplement 1), suggesting that regionally different phase responses are not directly derived from spatial ChrimsonR expression patterns. Given this distribution, and that we applied synchronous stimulation across the entire SCN, our results suggest that this subregional heterogeneity in resetting responses is intrinsic to the neurons or circuits in the ventrolateral and dorsomedial SCN.” (Results, pp. 9-10, lines 368-378)

6) For fold change data presented in Figures 2-4, comparison of differences between rising and falling durations becomes complicated with increasing light conditions. Since the authors are focusing on the change in symmetry of the waveforms, the authors can reanalyse the data as a ratio of the two durations – 'rising phase duration/falling phase duration' before and after stimulation instead of analysing each of the phases' duration separately.

We have previously considered showing the data as a ratio as the reviewer suggests. However, we decided not to present it in that way because it would be less informative than showing the data separately. For example, a ratio of less than 1 would occur in both a short T (the rising phase contraction) and a long T cycle (the falling phase elongation). By showing the data separately, readers can see how each phase of the cycle is changed differentially across these T cycles.

8) Page 7: "Optogenetic stimulation…. advances of about 4hr, respectively (Figure 2C)" There is considerable variation (upto 2.5 h) in phase shifts between animals. Instead of presenting the data as 'no phase shift' and 'about 4 h', please present the data as mean + sem here and in other places as well.

Thanks for your suggestion. We now have presented the data as mean ± sem there and in other places in the manuscript. The source data which will be presented in the supplement upon publication will also provide the readers with direct access to all the quantified data in detail.

Reviewer #3:In this study, the authors used long-term organotypic explant culture, cyclic red light optogenetic stimulation, and the PER2 bioluminescent reporter, to assess how the clock gene rhythms in the ex vivo SCN change in real time to achieve entrainment to light cycles. The use of the red wavelengths to drive optogenetic stimulation is an important advance as the blue light illumination common used produces phototoxicity in culture. The great strength of this work is that the authors provide data which enables us to visualize how the SCN clock responds to some of the classic environmental manipulations. To some readers, the weakness would be that the authors did not use their preparation to look at mechanistic questions. Still, in balance, the data showing how the hands of the clock are actually moving in response to classic entrainment protocols used for behavioural analysis is a major advance.

We greatly thank for the positive appreciation of our work.

Figure 1 shows the methodology used and provides compelling data on the benefits of the red light stimulation. Maybe this data could be placed in supplemental information as it is really about the methodology. Still it does nicely illustrate the costs of the commonly used blue light stimulation.In Figure 2, the show that the optogenetic stimulation alters the waveform of SCN PER2 rhythm following the classic phase response curve. With single 15min 10Hz optogenetic stimulation of SCN slices producing phase delays (CT 16), advances (CT21), and no phase shifts (CT 6). This data represents an important control and demonstrates that their system mimics the effects of light.To me, Figure 3 is the beginning off novel experiments. The authors used their optogenetic system to entrain PER2::LUC Rhythms in the SCN to different cycle lengths (22 and 25 hrs). I believe that this is a novel set of experiments and shows how the SCN waveform is altered to synchronize to different T-cycles.There is a long history of using skeleton photoperiods to mimic short (8 hrs per day), equinox (LD 12:12), or long (16 hrs per day). The authors used their optogenetic system to entrain PER2::LUC Rhythms to these different photoperiods. Entrainment to different skeleton photoperiods altered the molecular waveform of the SCN clock (Figure 4). Stimulations at dawn shortened the PER2::LUC rising phase, while stimulations at dusk lengthened the falling phase and set the phase angle of entrainment. Of course, these findings are a beautiful visualization of conceptual principles laid out by Pittendrigh and Daan. With Figure 4, there needs to be a little clarification to the graphs. We know that the authors are using 2 red light treatments to mimic dawn and dusk. So why three red lines? This was confuse when I was looking at the figure.

We apologize for the confusion. We think three red lines you are talking about are the ones in the 16:8 light cycle in Figure 4B. The very first stimulation occurred near the end of the day (2 hours before the trough) as described in Figure 4A, so this makes two dawn and one dusk stimulations shown on the second day of stimulation (or the fifth day on the actogram). We have changed the line color to yellow and grey for dawn and dusk pulses, respectively, to make them easier to distinguish.

It is well appreciated that the SCN is made of a number of cell types and usefully divided at the network level by VIP+ and AVP+ cell populations. The authors then assessed how the circadian phase shifts by single light pulses impact the network state of the SCN clock. They combined optogenetic stimulation of the SCN with spatially imaging real-time PER2::LUC bioluminescence in using a microscope to provide regional information. Following CT14 stimulation, the authors report that the lateral region showed larger phase delays than did the medial region. In contrast, period lengthening effects were more prominent in the medial SCN than in the lateral SCN. Following CT21 stimulation, the authors find the ventral SCN showed larger phase advances and smaller period shortening than did the dorsomedial or dorsal SCN. So that phase shifting data indicate an inverse correlation in magnitude between the phase shift and the period change among SCN subregions. Cluster analysis also identified a ventrolateral-dorsomedial axis for phase shifts and a ventral-dorsal axis for period changes. I believe that the reader could have used little more help with understanding the analysis of these experiments.

Thanks for the suggestion. We have now added a description of clustering analysis for general readers:

“To classify the SCN into subregions with different rhythmic properties in an unsupervised manner, we performed group-averaged clustering analysis and identified a lateral-medial axis with 3 clusters for phase shifts…” (Results, p. 9, lines 341-343)

Overall, the use of red optogenetic stimulation provides a technical advance over the blue wavelengths that are commonly used due to toxicity.The authors make use of this technology to illustrate beautifully how the SCN clock response to phase shifting stimuli, T- cycles, and skeleton photoperiods. Even a phenomenon as esoteric as phase jumps is reproduced in their cultures. The authors go on to demonstrate regional differences in how the SCN cell populations respond to phase shifts.Many of the findings are predicted by prior conceptual work in the field. But these prediction had not been put to the test so clearly before. To me this is a strength of the work.

Thanks for a positive evaluation of our work.

The writing and figures are very clear. Some jargon needs to be cleaned up. For example, "transforms the waveform…to highly asymmetrical trajectories" or "regional nodes" sound more suited to a modelling study than empirical work.

We thank for your suggestions. We have changed the text to “changes the waveform … to highly asymmetrical shapes.” (Abstract, p. 1, lines 16-18) We have changed the title to “Light sets the brain’s daily clock by regional quickening and slowing of the molecular clockworks at dawn and dusk.”

As I mentioned above, there may be too many red lines in some of their figures! The analysis of the data shown in Figure 6 needs a little more explanation for a general audience.

We changed the red lines to yellow and grey lines as described above. We have given more explanation for clustering analysis as described above.

The authors should at least discuss the possible impact of age. As I understand it, these cultures are P12 which is early in development.

Thanks for pointing this out. P12 is around the time when mice begin to open their eyes. By around P10, the mouse SCN gets innervated by retinal projections to a similar degree as the adult SCN, most clock genes become rhythmic, most SCN neuropeptides appear, and light responses reach near-adult levels (Bedont, Blackshaw, PMC4424844). To be fair, however, P12 SCN is not fully mature as the SCN astrocytes, a recently identified, important component for entrainment, are not fully mature by P20-P25 (Bedont, Blackshaw, PMC4424844). Nevertheless, our choice of P12 SCN is a sweet spot for our experiments that require a long survival and good amplitude rhythms ex vivo. We now have added in the discussion:

“We used ~P12 SCN slices to achieve long-term monitoring of real-time clock gene rhythms throughout optogenetic entrainment. Although the SCN astrocytes become fully mature by P20-25 and they are recently identified important component for entrainment in vivo, the SCN maturity reaches near-adult levels at P12 in many aspects including retinal innervation, clock gene rhythmicity, neuropeptide expression profile, and photic responses (Bedont and Blackshaw, 2015).” (Discussion, pp. 12-13, lines 510-515)

Also, is the sex of the animals mentioned?

Thanks for asking. We now have added “Both male and female mice were used in experiments.” (Methods, p. 15, lines 545-546)[Editors' note: further revisions were suggested prior to acceptance, as described below.]

Essential revisions:Thanks a lot, Dr. Kim and Dr. McMahon, for this revised version that has successfully addressed most of the points that were previously raised. All three reviewers appreciated the modifications in the text that have helped clarify key aspects of the work and made it more accessible to a broader audience. As suggested by Rev#2, there are two aspects that still need further clarification /editing.1) It is recommended to reword parts of the discussion relative to the comparison of in vivo ex vivo data, particularly when it comes to establishing possible causality. Whereas, the presented dataset is beautiful and establishes a nice correlation, yet there may be more intricate mechanisms in in vivo conditions.

We understand concerns over establishing causality in vivo from our ex vivo work. We have now revised the specific sentences in our manuscript identified by reviewer 2, and others, to make more clear how we delimit our conclusions, as shown below (indicate section, page, and lines for each):

“Daily waveform changes arise under ex vivo entrainment to simulated winter and summer photoperiods, and to non-24h periods. Ex vivo SCN imaging further suggests that acute waveform shifts are greatest in the ventrolateral SCN, while period effects are greatest in the dorsomedial SCN.” (Abstract, p.1, lines 20-23)

“Here we uncover that PER2 rhythms in the ex vivo SCN under entrainment to optogenetic light cycles show contraction of the rising phase and elongation of the falling phase depending on the timing of light exposure, and reveal ex vivo SCN plasticity at the clock gene level similar to canonical features of light-induced plasticity in circadian behavior. Aspects of circadian plasticity to light entrainment and regional heterogeneity of light responsiveness are apparently intrinsic to the SCN clockworks.” (Introduction, p. 3, lines 84-90)

“However, our results from skeleton photoperiods suggest that light-dark transitions at dawn and dusk are sufficient to alter PER2 waveform width.” (Discussion, p. 10, lines 438-439)

“As the VIP and AVP neurons are respectively located in the ventral and dorsal SCN, this suggests that regionally differential phase and period responses in the SCN might be derived from intrinsic differences between the VIP and AVP neuronal clocks.” (Discussion, p. 11, lines 500-503)

2) The manuscript still concludes that the ChrimonR-based method is less toxic for in vitro optogenetic perturbation of the SCN than others based on blue light. As indicated by Rev#2 this may be too strong of a conclusion, as "toxicity" was only indirectly inferred from luciferase arrhythmic signals and assessed only after prolonged light-stimulation (which is not utilized in skeleton photoperiods). While this could be addressed by additional experiments that would actually establish such cytotoxicity, it would be also possible to rephrase parts of this topic.Indeed, the data presented in the manuscript strongly suggests that prolonged stimulation with blue light is not an option for conducting full photoperiod circadian optogenetic studies. Yet, the dataset does not provide strong proof on whether this is due to cytotoxicity or another reason. Nevertheless, as the blue-light protocol tested in the manuscript (12 h pulse) is not directly comparable with the ones used for red-light throughout the paper (15 min pulses) this suggests that -potentially- one could have used a blue-light system and yet get all experiments to work just fine. But a valid, and relevant point is that Figure 1 shows that in a long optogenetic stimulation (12 h) the red-light systems outruns the blue-light one (albeit such stimulation protocol is not used later on throughout the paper, as the technical system does not allow one to stimulate and record at the same time). Thus, the data seem to strongly suggest that the ChrimsonR system has the "potential" of being more versatile as the quality of the rhythms are not affected by prolonged red-light stimulation, and would allow to test and compare (in future experiments) full photocycles (8:16, 12:12, 16:8) and skeleton ones, something that would be extremely difficult with blue-light.

We agree that our data do not make a strong case against using “brief” blue light exposure for circadian studies and we do not claim in our manuscript that optogenetic stimulation with blue light should be avoided. As pointed out by the editor, however, our data suggest that ChrimsonR system could be potentially more useful to simulate light paradigms consisting of prolonged light exposure that can impact or impair circadian rhythmicity in a brain slice. We have now changed the text accordingly shown below to tone down or replace strong terms such as photodynamic damage and better reflect our main points.

“This effect was not reversible with a medium change (Figure 1A), suggesting that long-term blue light exposure per se can impair circadian rhythmicity in SCN slice cultures.” (Results, p. 3, lines 106-108)

“Since side effects of light exposure decrease with increasing irradiation wavelength (Tyssowski and Gray, 2019; Waldchen et al., 2015), we tested whether using red light (625nm) mitigates light impairment of SCN rhythms.” (Results, p. 3, lines 109-111)

“12h red light pulses (625nm, 10Hz, 10ms, 1.25mW/mm2) did not significantly affect the PER2::LUC rhythm in SCN slices (Figure 1A, B), suggesting that using red light stimulation could be more feasible for prolonged optogenetic light stimulation ex vivo.” (Results, p. 3, lines 111-114)

“To extend the duration over which we can observe the SCN slice throughout entrainment, we improved the quality and robustness of PER2::LUC rhythmicity by using brain slices from younger mice that usually survive longer in culture (Humpel, 2015), and by using stabilized glutamine media that were shown to reduce ammonia production and improve cell viability in cell culture (Christie and Butler, 1999; Imamoto et al., 2013) (Figure 1 —figure supplement 1).” (Results, pp. 3-4, lines 119-124)

“Using real-time bioluminescent recording of PER2 expression combined with recurring optogenetic stimulation, …” (Discussion, p. 9, lines 387-388)

In addition, in revising the passage in the Results on our experimental conditions according to the reviewers comments, we also realized that we had left out a detail about the interval for media changes that could be key for the understanding of non-expert readers. Thus, we have edited that description to read:

“With these modifications, ex vivo SCN rhythms were stable for more than three weeks, long enough to conduct entrainment paradigms without culture medium changes that may perturb ex vivo SCN rhythms and entrainment.” (Results, p. 4, lines 124-126)

Reviewer #1:The manuscript has included important changes that help grasp the main findings of the work. Thus, it describes an ex vivo optogenetic experimental platform that allows precise optogenetic stimulation of explanted SCN slices while tracking clock gene expression with high temporal resolution. Importantly, the use of red-light does not bear some of the problems of cytotoxicity associated with blue-light and therefore allows multiple cycles of optogenetic modulation.The most critical points raised by reviewers were successfully addressed.

We thank again for your constructive review.

Reviewer #2:The authors are commended for their revisions to the text and analyses to accommodate comments from the editor and reviewers. They retain the same conclusions: the SCN in vitro entrains with many of the same features of circadian systems in vivo to different photoperiods. I recommend they reconsider their decisions on two major points they feel are important to the paper:1. The authors now include additional similarities between SCN entrainment in vitro to optogenetic activation pulses and in vivo entrainment to long photoperiods. They find parallels between skeleton photoperiods in vivo and in vitro (e.g. psi jump) and period after effects to single pulses in vivo and in vitro. These correlated changes in PER2 expression in the cultured SCN should not, however, be used as evidence for their sufficiency or necessity in photic entrainment in vivo. I recommend removing causal arguments such as. "Daily waveform changes are sufficient to entrain to simulated winter and summer photoperiods, and to non-24h periods. SCN imaging further reveals that acute waveform shifts are greatest in the ventrolateral SCN, while period effects are greatest in the dorsomedial SCN." (abstract) and "PER2 rhythms in the SCN entrain to external light cycles via contraction of the rising phase and elongation of the falling phase depending on the timing of light exposure, and show how canonical features of light-induced plasticity in circadian behavior are expressed at the clock gene level. Circadian behavioral plasticity and regional heterogeneity of light responsiveness are intrinsic to the SCN clockworks." (Introduction).

Thanks for your suggestion. We have now toned-down causality statements throughout the manuscript as shown above in responses to the editor.

2. They conclude they have developed a method that is less toxic for in vitro optogenetic perturbation of the SCN. Two reviewers challenged the authors to be more quantitative and thorough before concluding that the use of ChrimsonR and their modified culture media offer improvements over prior methods. The authors dedicate Figure 1 and Supplementary Figures 1 and 2 to this point. Figure 1 clearly shows that 12h of blue light dramatically decreased the bioluminescent reporter and a medium change did not rescue expression in 1-3 examples. Suppl Figure 1 provides a single comparison of young SCN + modified media to an adult SCN without modified. The authors use strong terms like "phototoxicity" and "photodynamic damage" and "build-up of toxic ammonia" without showing any damage to the SCN beyond loss of bioluminescence following 12 h exposure to blue light (a treatment they do not use for any of the subsequent experiments in the paper). They may be right that 12 h of blue light (10 Hz, 10 ms flashes) is incompatible with studies that aim to activate the SCN in vitro. However, we need more information about the intensity of the blue light used and whether dimmer light could be used to activate ChR with results similar to ChrimsonR. To argue ChrimsonR offers advantages over ChR, the paper would benefit from more than three replicates of the 12 h effect and, most relevant to this paper, evidence that the authors cannot measure the changes in PER2::Luc with skeleton photoperiods applied to ChR. As presented, the authors are welcome to justify their choice of media and CrimsonR, but they should not encourage researchers to change their media and optogenetic probe to avoid toxic side-effects.

We understand the reviewer’s points. We agree that our data provide indirect evidence that blue light exposure itself can elicit side effects in culture though the effect magnitude may be subject to culture conditions or light intensities. However, we do not claim that optogenetic stimulation with blue light should be avoided for all circadian rhythm experiments. To be fair, our data address general concerns raised in other fields over side effects of blue light exposure in culture, and show that ChrimsonR system is an alternative option to utilize optogenetics for circadian rhythm studies. We have now changed the text accordingly as shown below to tone down or replace strong terms such as photodynamic damage and better reflect our main points as shown in the response to the editor above.